# Decoupling of timescales reveals sparse convergent CPG network in the adult spinal cord

Marija Radosevic [1], Alex Willumsen [1], Peter C. Petersen [1,2], Henrik Lindén [1], Mikkel Vestergaard [1,3] & Rune W. Berg [1]

During the generation of rhythmic movements, most spinal neurons receive an oscillatory synaptic drive. The neuronal architecture underlying this drive is unknown, and the corresponding network size and sparseness have not yet been addressed. If the input originates from a small central pattern generator (CPG) with dense divergent connectivity, it will induce correlated input to all receiving neurons, while sparse convergent wiring will induce a weak correlation, if any. Here, we use pairwise recordings of spinal neurons to measure synaptic correlations and thus infer the wiring architecture qualitatively. A strong correlation on a slow timescale implies functional relatedness and a common source, which will also cause correlation on fast timescale due to shared synaptic connections. However, we consistently find marginal coupling between slow and fast correlations regardless of neuronal identity. This suggests either sparse convergent connectivity or a CPG network with recurrent inhibition that actively decorrelates common input.

[1] Department of Neuroscience, Faculty of Health and Medical Sciences, University of Copenhagen, Blegdamsvej 3, DK-2200 Copenhagen N, Denmark. [2] Present address: Neuroscience Institute, New York University, New York, NY 10016, USA. [3] Present address: Department of Neuroscience, Max Delbrück Center for Molecular Medicine (MDC), 13125 Berlin-Buch, Germany. Correspondence and requests for materials should be addressed to R.W.B. (email: runeb@sund.ku.dk)

Movement is an essential part of our daily lives, and disorders of the motor system, such as spasticity, amyotrophic lateral sclerosis, and spinal cord injury are particularly debilitating for individuals. Simple rhythmic movements, such as walking and breathing, have constituted models for fundamental aspects of the motor system. In spite of extensive investigations[1–6], the connectivity of the network responsible for generating the motor activity remains unknown. A circuit component, known as a central pattern generator (CPG), is believed to transmit command signals to motoneurons and local premotor interneurons[7–10]. Although the size of the respiratory motor network, i.e. the preBötzinger complex[1,11], is well-known, the size and wiring of other CPG networks are not well understood. A feedforward organization is often proposed between groups of neurons or modules, which exhibit alternating rhythmic bursting[12] (Fig. 1a). Common drive modules are thought to be small, e.g., the preBötzinger complex has only 600 neurons[1], which provides rhythmic drive for the rest of the network. The projection is also believed to diverge onto a much larger population of receiving neurons[13–15]. Thus, the receiving neurons would share the same connections via a dense divergent connectivity (Fig. 1b). Since the transmission is communicated by action potentials, which are precise in time, a dense connectivity will manifest as a strong temporal correlation between synaptic potentials in the receiving neurons, and this correlation can be verified experimentally through pairwise recordings. If the drive network is not a small but rather a large population, however, the receiver neurons are likely to collect sparse convergent input without correlation (Fig. 1c). Hence, the assessment of correlation via pairwise sampling from local neurons will provide important information about the fundamental structure of the premotor network.[16–18]

Here, we use hindlimb scratching of adult turtles as a model for stereotypical rhythmic movement, and investigate the pairwise correlation between motoneurons, as well as interneurons, in a spinal cord network (see Supplementary Movie 1 for a video abstract). The turtle preparation offers the unique advantage of being resistant to anoxia, which permits retaining functionally-intact motor activity induced purely by natural somatic stimuli. Further, the mechanical stability of this preparation allows remarkable access to synaptic input across pairs of neurons via dual intracellular recording[19,20]. First, we utilize dual intracellular recordings to assess the strength of synaptic correlations, in particular for pairs belonging to the same module. The modular–association is based on two issues: (1) motor neuron

pairs in close vicinity and with same slow phase (2) interneurons also in close vicinity that have same phase are assumed to belong to same module and receive common drive. This assumption is based on the consensus view that the common–drive network is small compared with the receiver network (hence the term 'common') and therefore the risk of randomly recording from one of the source–network neurons is equally small. Next, we use multi–electrode arrays to measure population activity to determine the pairwise spike–spike correlation as an additional indicator for shared synaptic input, under same assumptions. In both approaches, we found a consistent decoupling between the slow rate modulation and the fast synaptic activity, even for pairs belonging to the same module. This indicates that the similarity in slow rhythmic activity across spinal neurons is not due to input from the same source. We propose two explanations for this paradoxical observation. First, according to a minimalist feedforward model, the common drive network must be large with sparse convergent connections. Alternatively, the network does not have a pure feedforward architecture, but includes recurrent connectivity[21,22] and consequently *active decorrelation*. Active decorrelation is a mechanism observed, e.g., in the neocortex, by which correlated input due to shared connectivity is partially cancelled by inhibition[23–26]. The latter interpretation, if true, implies a role of inhibition in motor circuits, which is fundamentally different from the previously assumed role of inhibition in the spinal cord.

## Results

**Paradigm**. We recorded from pairs of spinal neurons ($n = 66$ pairs) either motor neurons (MNs) or interneurons (INs) located in the lumbar region of adult turtles (segments D9/D10). Multi-electrode arrays were also inserted into the same region of the spinal cord (up to 256 channels, in $n = 6$ animals) in order to investigate the correlated activity of a subset of the neuronal population[20,27]. A computational model was implemented to assist in predicting the strength of correlated input for different degrees of sparseness within the common source network.

**Correlation to infer sparseness and size: predictions**. The correlation of synaptic input across a pool of receiving neurons depends on how many shared connections they obtain from a source network. To quantify this, we employed a minimalist model, which consists of a source network of variable size that projects to a receiver network, which represents the local spinal

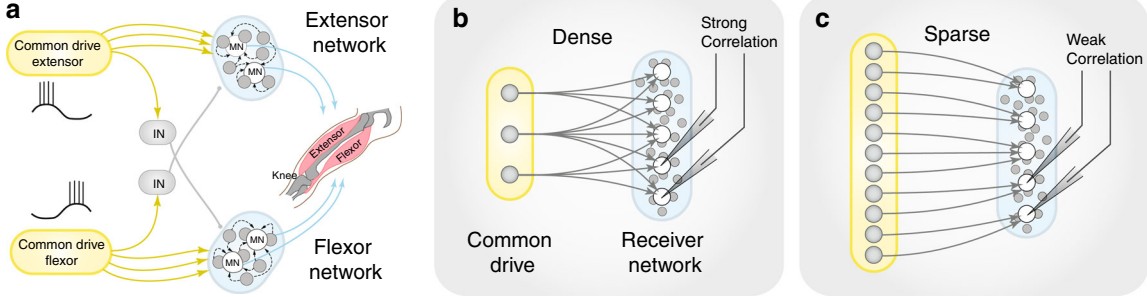

**Fig. 1** Scenarios of feedforward connectivity of motor networks and the expected pairwise correlation. **a** Traditional half-center model with feedforward connections from a common drive network (yellow shaded area) to flexor- and extensor-related neurons in the spinal cord (blue shaded region with local premotor neurons in gray) including reciprocal inhibition (IN). The common–drive network of unknown size and architecture projects to functionally related local neurons in the lumbar spinal cord (blue shaded region). **b** A densely connected input from the common–drive network consisting of few neurons with many connections per neuron (high *out-degree*) results in strongly correlated input across the receiver neurons (blue). **c** A sparsely connected and large common–drive network with small out-degree results in weak correlation across the receiver neurons. The number of connections coming into the receiver network (the *in-degree* distribution) is the same as in **b**

INs and MNs (Fig. 1). A given number of arriving synaptic connections to a group of neurons can either be provided by a small population of neurons (yellow) with many axon collaterals (Fig. 1b), i.e., a dense/divergent connectivity, or a large population (yellow) with few axon collaterals, i.e., a sparse/convergent connectivity (Fig. 1c). Thus, synaptic correlation in the receiver network can provide important insight about the connectivity, i.e., whether it is dense or sparse. In graph theory, density is often defined as the number of connections ($k$) divided by the total possible number of connections[28,29] ($n$), i.e. $k/n$ (see methods). We define sparseness ($\rho$) as the inverse of density, i.e. $\rho = 1 - k/n$. Since the correlation of input across two receiving neurons does not depend on how many other neurons receive the same input, the size of the receiver network is irrelevant and therefore kept constant. Consequently, there will be a large overlap in input, and thus high correlation when the source network is small (dense connectivity). This is opposed to when the network is large with convergent connections, in which the correlation is expected to be negligible.

Testing this prediction in our model, we found that a dense input from a small group of common drive neurons ($\rho = 0.5$) with rhythmic yet independently Poisson spiking (Fig. 2a) caused a high correlation in membrane potential of pairs of neurons. In the contrary situation, a large and sparsely connected common drive network ($\rho = 0.98$) evoked membrane potential fluctuations across pairs of receiving neurons, which had little resemblance other than the slow rhythm (Fig. 2b). The distribution of synaptic correlation between pairs was relatively high for the dense/divergent network (top, Fig. 2c) while the sparse/convergent

network had a near–zero value (bottom). Both distributions exhibited a large variance around the mean ($\star$). Such large variance around a small mean for the sparse architecture is qualitatively similar to what has been observed in balanced neocortical networks[23,24,30]. Variability can also be seen in the correlation matrix (right). Changing the architecture in our model from dense to sparse, we observed a direct inverse relation between sparseness and the correlation of synaptic input in the receiving cells (Fig. 2d). Further, the mean correlation coefficient was dependent on the size of the source network. The correlation showed a graceful decay with network size as $1/n$ (Fig. 2e). The network sparseness, on the other hand, climbed towards 1 as the size of the network increased (gray line). In conclusion, the correlation between input to a pair of randomly selected receiving neurons is an indicator of both network sparseness and the relative size of the common source network. Specifically, the weaker the shared input is, the more large and sparse is the source network.

**Functional modules grouped by phase.** The premise of the above analysis is that a given pair of receiving neurons belongs to the same functional module. However, a local spinal population consists of neurons involved in various activities from flexion to extension, as well as other synergist contractions with various phase lags[31]. These distinct groups of neurons are naturally expected to receive uncorrelated input due to differences in origin. To determine the sources, we distinguish between pairs of neurons according to their phase. Specifically, those with zero phase lag belong to the same functional module[3,32], while those

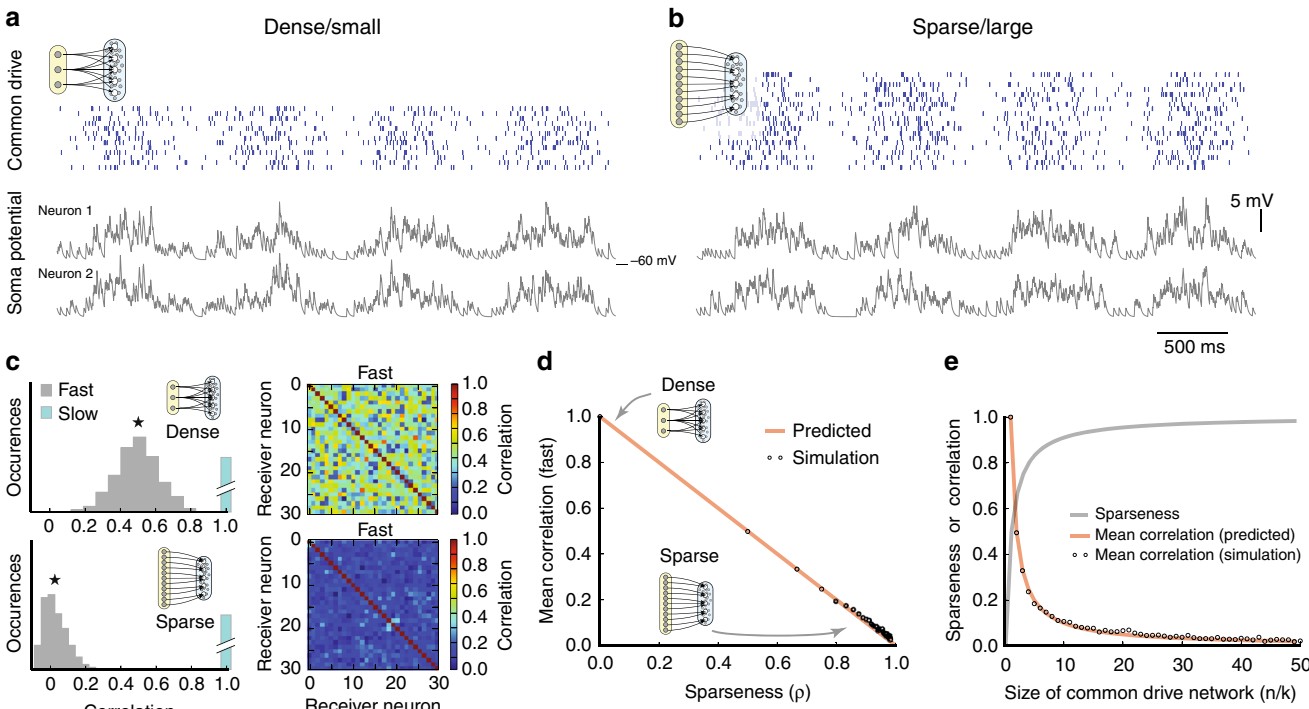

**Fig. 2** Synaptic correlation depends on network size and architecture. **a** Oscillatory spiking activity of the motor rhythm was imitated as an inhomogeneous poisson process with sinusoidal rate in a small common drive model network (raster) projecting via divergent connections results in a rhythmic $V_m$ of target neurons with substantial correlation on both fast and slow timescales (below/above 400ms–timescale). Sample traces shown (gray), $R = 0.5$ on fast timescale. **b** Sparse convergent connectivity from a larger premotor network gives weak correlation ($R = 0.02$, fast). **c** Distribution of fast correlation for receiving pairs with dense/small (top) versus sparse/large connectivity (bottom) for the conditions in a ($n = 20$ source cells) and b ($n = 500$). Correlation matrix for the receiving population shown in colors to the right. The slow correlation remains 1 in both topologies (cyan). **d** Mean fast correlation (star in c) is inversely linked to sparseness (circles). The expected correlation (orange, 'predicted') is the fraction of common connections over the total number of synaptic input. **e** Sparseness (gray) increases and fast correlation ('predicted', orange, 'simulated' circles, as in d) decreases with network size. The in–degree is kept constant ($k = 10$)

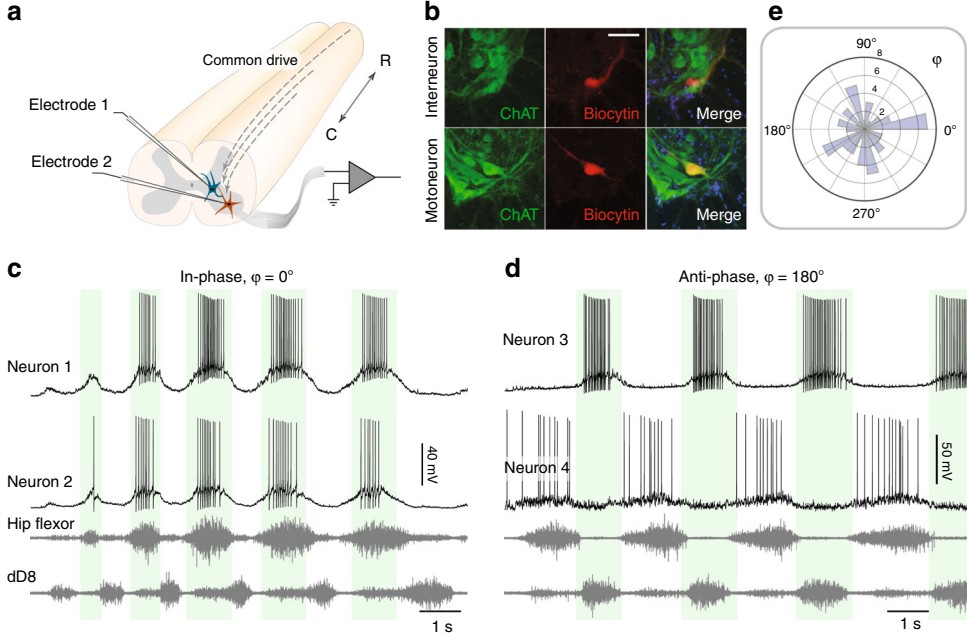

**Fig. 3** Diverse phase relations during spinal motor patterns. **a** Experimental setup: Pairwise intracellular recording from the lumbar spinal cord of turtles (medial-ventral horn) during touch-induced scratching. **b** Histology of a recorded pair of neurons filled with biocytin (red) reveals an IN (Choline Acetyl Transferase (ChAT)-negative, top row) or a MN (ChAT-positive neuron, green, bottom row). Scale bar 50 $\mu m$. **c** Two in-phase neurons during motor activity concurrent with hip-flexor and a knee-extensor synergist (dD8) nerve activity. **d** Anti-phase MN-IN neuron pair (histology shown in from **b**). Shaded regions indicate on-phase of the top panel neuron. (**e**) Phase between pairs indicates wide representation from 0 to 360° ($n = 66$ pairs). *ChAT* choline acetyltransferase

with large phase lags are incompatible. Hence, we expect to observe an increase in correlation in fast synaptic input as the correlation in slow rate modulation increases, i.e., a coupling between timescales.

To verify a coupling between slow and fast correlations, we first conducted dual intracellular recordings from both MNs and INs in the lumbar spinal cord of turtles performing touch-induced scratching (Fig. 3a). To avoid the confounding factor of supraspinal input, the interference with and prevention of the scratching response to sensory stimulation, the turtles were spinalized. The muscles were removed to limit proprioceptive feedback and increase mechanical stability, while leaving the cutaneous sensation intact. Post-hoc immunohistology and filling with an intracellular marker (biocytin, red) was performed to assist in the cell-type identification (Fig. 3b). Pairwise recordings showed neurons that had a slow rhythmic activity concurrent with a particular nerve, i.e., in-phase activity (Fig. 3c). Here, neuron 1 and 2 were in-phase with each other ($\varphi = 0$), as well as the hip-flexor nerve activity. Another sampled pair exhibited anti-phase activity ($\varphi = 180°$) in spite of their close proximity (Fig. 3d). In fact, all of the neuronal pairs had commensurate oscillations with various phase lags (Fig. 3e) regardless of their close physical location (<300 $\mu m$). A similar poor relationship between physical location and phase delay has been previously observed in turtles[33], as well as other animals, e.g., neonatal mice[4,34,35]. Consequently, spinal neurons receive various common drives, and it is not possible to rely on their somata location for classification of functional relatedness.

**Slow and fast synaptic activities are decoupled**. Instead of using somata location to classify functional relatedness between neuronal pairs, we utilize the cross-correlation of their rhythmic activity. If a pair is in-phase, both neurons will presumably belong to the same flexor/extensor module[31,32]. If they belong to the same module, they should receive a large proportion of

synaptic input from the same source, and this will manifest as a strong correlation on a fast synaptic timescale. To quantify this, we divided the membrane potential (Fig. 4a) into slow and fast signals (Fig. 4b). Correlation in the slow signal represents functional relatedness; whereas, correlation in the fast signal represents directly shared synaptic input. The fast signals were correlated in a sliding 400ms-window in order to probe temporal aspects of the correlation (Fig. 4c, d). Even though the correlation on a fast timescale exhibited some variability, it did not possess a clear relationship with motor rhythm (blue line). The variability can be associated with uncertainty in estimation since the shuffled data contained similar dynamics (blue and beige lines Fig. 4d). Although this sample pair had the highest correlation on a fast timescale among all pairs, the correlation was rather weak when comparing the distribution of correlations over time with the shuffled data (vertical histograms Fig. 4d). For the fast correlation averaged over the whole trace (red, Fig. 4e), this sample pair exhibited an almost perfect correlation on a slow timescale (gray, Fig. 4e) yet a relatively weak synaptic correlation (compare 0.36 with 0.99). Thus, we conclude that the strong correlation on a slow timescale was not caused by shared synaptic input.

This absence of fast correlation could be due to electrotonic filtering within the cell if the synaptic contacts of the correlated source were located far apart on the cell. To test this, we performed dual recordings ($n = 5$) of the same neuron (not the one in a–d). We recorded the membrane potential of the same neuron at two locations of the cell (Fig. 4f). Here, an electrotonic separation of the potentials should manifest as a lack of pairwise correlation. Nevertheless, the correlation in synaptic potentials of the two electrodes were close to 1 throughout the activity (Fig. 4g) and much higher than for the correlation between the two different neurons (Fig. 4e). A corresponding high correlation was also found in the slow signal (Fig. 4h). Similar results were found in the other pairwise recordings ($n = 5$). Thus, we conclude that the lack of correlation on a fast timescale between two neurons is

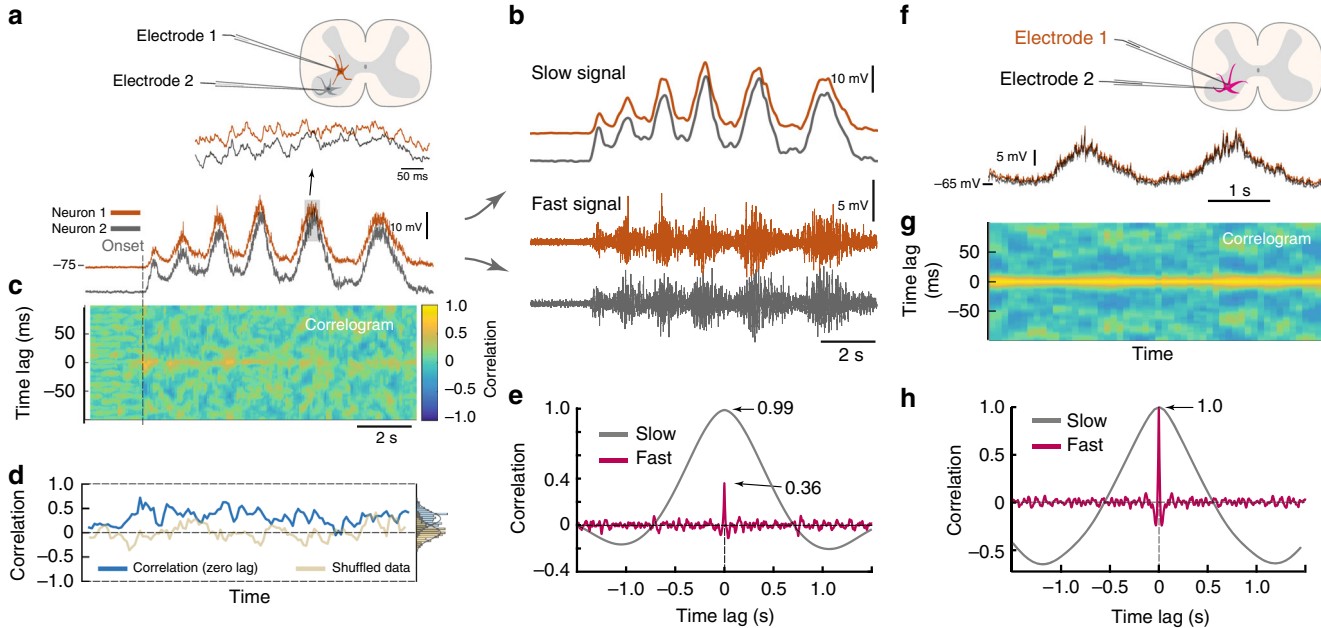

**Fig. 4** Two timescales of synaptic input. **a** Sample data of recordings of two spinal neurons kept hyperpolarized to prevent spiking during network activity (onset indicated, pair from Fig. 3c). Highlighted region indicates concurrent membrane potentials. **b** Separating traces into slow (top) and fast signals (bottom) by digital filtering. **c** Common synaptic input is quantified by the correlation of the fast signal in a temporal window (400 ms) moving in time, i.e. correlogram with time on x-axis, time-lag on y-axis. A small albeit significant correlation is seen as a yellow horizontal shadow. **d** The zero-lag correlation (blue) indicates no clear temporal relation to the rhythm. Distribution shown vertically (right, $\mu = 0.29$). Shuffled data (beige). **e** Strong slow correlation (0.99), whereas the fast is weak (0.36, excluding quiescence), indicating a decoupling between slow rate modulation and fast synaptic input. **f** Paired recording from same cell as control. **g** Correlogram for same-cell dual recordings similar to **c**. **h** Same-cell recording show coupling between slow and fast correlation (both peak at 1)

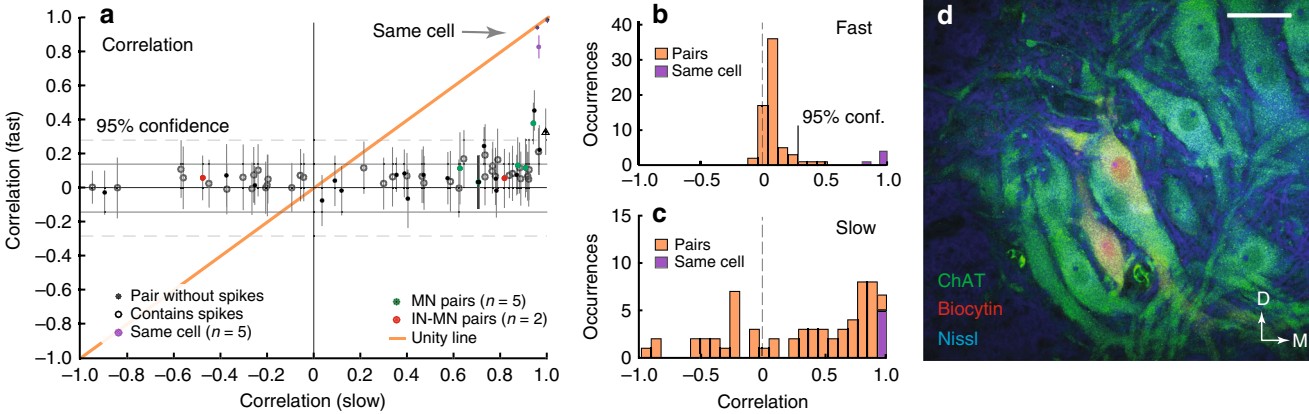

**Fig. 5** Decoupling of slow and fast timescales of synaptic input. **a** Correlation in synaptic input to pairs of neurons ($n = 66$) on slow timescale (i.e. input rate modulation, abscissa) vs. fast correlation (synaptic potentials, ordinate). A departure from the unity line (orange) indicates that correlated rate modulation is not caused by the same synaptic drive. As a control, paired recordings from same cells ($n = 5$) do not exhibit such a departure (purple points indicated). Confidence limits assessed as $2\sigma$ from the mean of shuffled data ($\sigma$ = standard deviation). Pair from Fig. 4a–e indicated (triangle). Error bars indicate standard deviation. **b** Synaptic correlation (fast) is scattered close to zero (with a slight positive bias) and has only 4.5% ($n = 3/66$) above the 95% confidence limit. **c** Slow correlation has a weak mode close to 1. **d** Close proximity of MNs (biocytin (red) and Choline Acetyl Transferase–stained (ChAT, green)) does not grant high correlation in synaptic input ($R = 0.36$, right green point in **a**). Nissl stain in blue. Location in the left ventral horn. Scale bar 50 $\mu m$. D: Dorsal, M: Medial. Each point in **a** represents the mean correlation of repeated measures of same sample pair with standard deviation as error bars

not due to electrotonic filtering. These data also ensure that the recorded fluctuations, which may appear to be noise, are correlated in both electrodes, and it is therefore not electronic noise, but rather fluctuations in synaptic input.

Separation of fast and slow synaptic timescales was performed for all pairs, both IN-MN and MN–MN pairs ($n = 66$ in total), and plotted against each other (Fig. 5a). Pairs that belong to the same functional module, i.e., have a strong slow correlation, did

not exhibit a parallel correlation in the fast activity. This was shown as a departure from the unity line, i.e., a decoupling of timescales (Fig. 5a). A majority of the pairs had a near zero correlation on a fast timescale (Fig. 5b). Only three out of 66 pairs (4.5%) had a correlation higher than $2\sigma$ from the mean of the distribution for shuffled pairs, which is within what is expected by chance for the 95% confidence limit (solid vertical line). When choosing slow correlation >0.8 there was $n = 3/17$ points (17.6%)

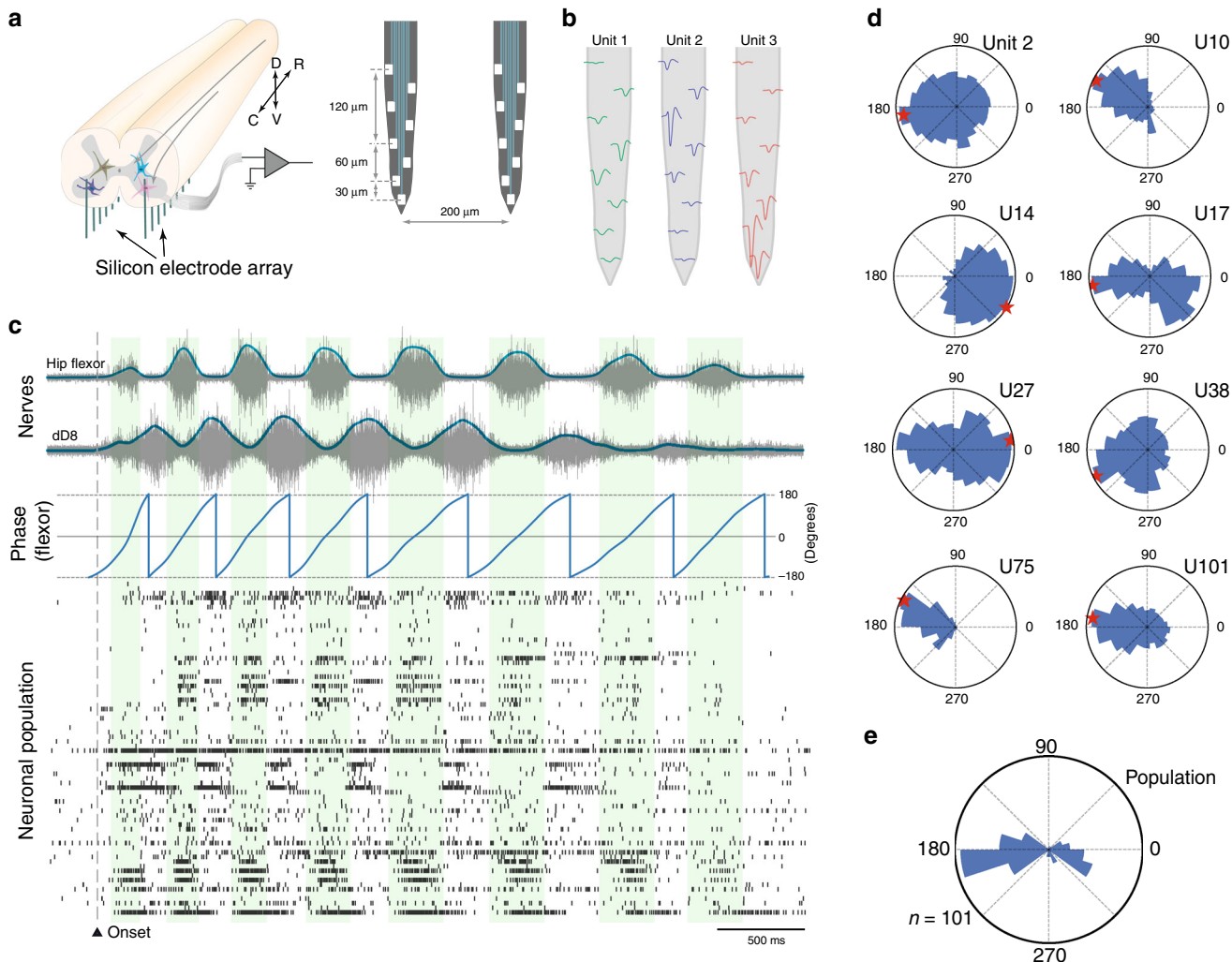

**Fig. 6** Population spiking of lumbar neurons. **a** Silicon multi-electrode arrays (right) were inserted ventrally into the medial–ventral horn in the lumbar spinal cord (left) of semi-intact preparation. **b** Three sample units and their waveforms recorded on the different shank electrode. **c** Touch-induced scratching (onset) has alternating hip flexor (blue top) and knee extensor (blue bottom, dD8). Filtered and Hilbert-transformed motor nerve as a phase reference of the rhythm (blue). Raster plot: The spiking activity of lumbar neurons ($n = 72$) has scattered rhythmic activity. **d** Spike-triggered phase histogram in polar plot (blue in **c**), for 8 sample neurons (with various unit numbers), show preferred phase (red star) as well as diversity in spiking. **e** Histogram of preferred phases (red stars in **d**) for the entire population ($n = 72$)

above confidence limit. Nevertheless, the fast correlation in this group was not statistically different from the group with slow correlation below 0.8 (Supplementary Fig. 1). For comparison, no decoupling was observed for dual recordings from the same cell ($n = 5$, purple Fig. 5a–c). The correlation on a slow timescale had a scattered distribution from negative to positive values (Fig. 5c) as expected, considering the previously observed phase distribution (Fig. 3e). Even pairs within close proximity (Fig. 5d) exhibited a remarkable decoupling (0.99 vs. 0.36, green point Fig. 5a). This indicates a pervasive decorrelation of synaptic input among pairs of neurons of the same functional module. Based on our previous analysis (Fig. 2d) and since the fast correlation is below the significance level (0.0–0.2) the network sparseness is probably about 90% ($\rho \sim 0.90$). Although we neither know the size of common drive network ($n$), nor the in–degree ($k$), the feedforward connectivity would have to be sparse, i.e. large $n/k$, in order to achieve such low correlation on fast timescale (Fig. 2e).

**Spike-spike correlations are decoupled from rate modulation.** Until now, our analysis has focused on common source input to

pairs of neurons in the lumbar cord, which demonstrated a decoupling between synchrony in slow and fast synaptic activity. Since the correlated synaptic input often causes synchronized discharge[36] we can further substantiate the decoupling by investigating the concurrent spiking of the IN and MN population. To achieve this, we inserted multi-electrode arrays (128–256 channels) into the same part of the lumbar region to greatly increase the number of neuronal pairs in our analysis (Fig. 6a). Single units were sorted using polytrode spike sorting[19] (Fig. 6b). This gave the concurrent spike activity of typically ~300 neurons in addition to multiple nerve recordings (Fig. 6c). Similar to the intracellular data, different units had different phase–preference in rate modulation. The phase of the hip flexor activity (from Hilbert–transformation, third panel) was used to characterize the diversity of the population. The spike–triggered phase distribution was calculated for each unit and plotted in polar histograms, which illustrates the high degree of complexity in the population activity (Fig. 6d–e).

To perform a similar analysis to the pairwise intracellular data (Figs. 4 and 5), the spike activity of individual neurons was divided into a fast and a slow component by convolving the spike

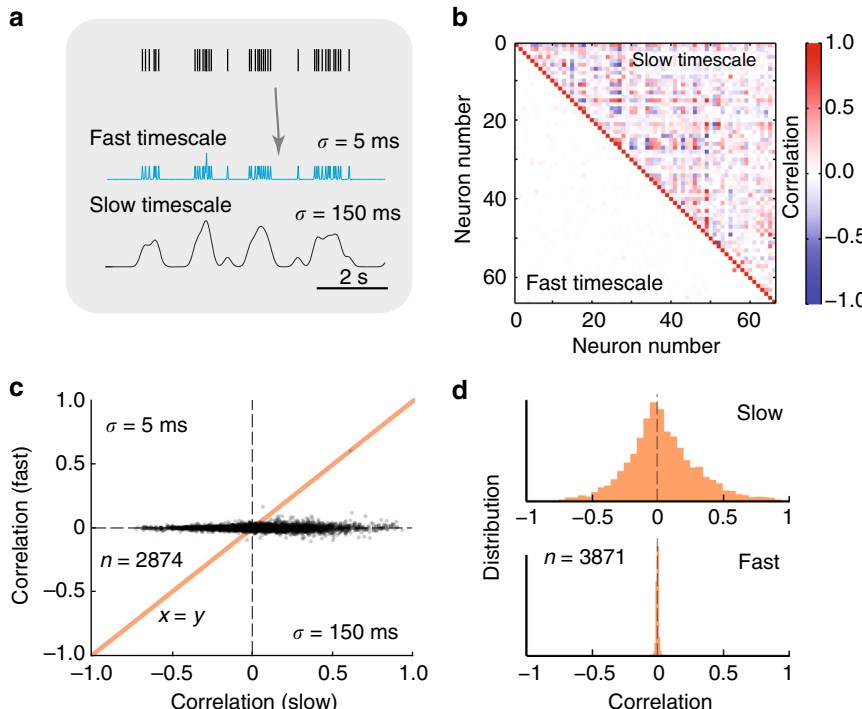

**Fig. 7** Decoupling of slow and fast-spiking timescales. **a** Spike trace separated into slow and fast activity by convolving spike times with different Gauss–kernels ($\sigma = 5$ and 150 ms). **b** Correlation matrix for the neuronal spiking on fast (lower left) and slow timescales (upper right) indicate a decoupling between the functional relation across pairs of neurons and their synaptic input. **c** Pairwise correlation on slow (x-axis) and fast timescales (y-axis) indicate a decoupling and departure from the unity line (orange). $n = 2874$ pairs, two hemicords. **d** Histogram of slow correlation (top) is broadly scattered whereas the fast correlation is scattered tightly around zero (bottom)

times with a narrow and a broad Gaussian kernel, $\sigma = 5$ and 150 ms, respectively (Fig. 7a). These traces were then correlated for all neuronal pairs, and the coefficients were plotted in a composite correlation matrix in colors (Fig. 7b). The upper–right half shows the pairwise correlation in slow rate modulation, which contained large negative-to-positive values (blue to red). The lower–left half shows the pairwise correlation on a fast timescale. The white appearance indicates a near–zero value for all pairs. To directly test the coupling between fast and slow correlations, all correlation values were plotted against each other (Fig. 7c). A distinct departure from the unity line (orange) was observed, similar to the intracellular data (Fig. 5a). The distribution of correlation coefficient on a fast timescale was thinly scattered around zero (bottom, Fig. 7d) indicating an absence of correlated spiking, even though many pairs had a strong slow correlation (top) and were therefore functionally related (see Supplementary Discussion on discrepancy between intra– and extracellular data). This decoupling between timescales is an indication for the lack of correlated synaptic input, in particular for pairs that belong to the same module, and therefore should have received common input.

**Local connectivity is sparse**. So far, our analysis has focused on shared input to neighboring neurons or correlations in spiking activity of extracellularly–identified neurons, without considering local recurrent connectivity especially inhibitory connection, which are important for active decorrelation. By combining intracellular and extracellular recordings together within the lumbar region, we were able to identify local connections and directly test whether they are inhibitory or excitatory. By inserting an intracellular electrode in conjunction with the Si-arrays

(Fig. 8a) we compared the timing of spikes by neurons in the population with synaptic events in the membrane potential ($V_m$). The $V_m$ of one neuron (Fig. 8b) is shown together with the population spiking activity (Fig. 8c). If there was a connection from an extracellularly-recorded neuron to the intracellularly recorded neuron, a spike should evoke either an excitatory or an inhibitory post-synaptic potential (EPSP or IPSP). Consequently, we could verify both a connection and its identity. An inhibitory connection was confirmed by a spike–triggered median trace with a negative peak significantly outside of the reference distribution (Fig. 8d). The associated extracellular waveform of the inhibitory cell was recorded on multiple electrodes on the shank (Fig. 8e). The decay–time constant of ~5 ms of the identified inhibitory cells ($n = 5$) was comparable to previously identified glycinergic inhibition[37,38] (Fig. 8f). The significance of the synaptic connections was established by comparison of the IPSP–peak with that of a surrogate data, where a temporal structure had been abolished by random jitter of the spike–times. The PSP peak distribution for a data set containing 4 such inhibitory units is shown (Fig. 8g). The corresponding spike activity for the same data set containing the 4 inhibitory activity is shown in blue (Fig. 8c), where one inhibitory unit is concurrently active with excitation, while three are alternating, indicating the complexity of the population activity[39]. The verified synaptic connections give a connectivity probability of only 0.9% of inhibitory input, and even lower for the excitatory units (none detected). Similar low estimates (1%) has previously been obtained indirectly in the medulla respiratory system[11,40].

**Decorrelation by recurrent inhibition**. The observation that the local neurons we recorded from were all inhibitory neurons

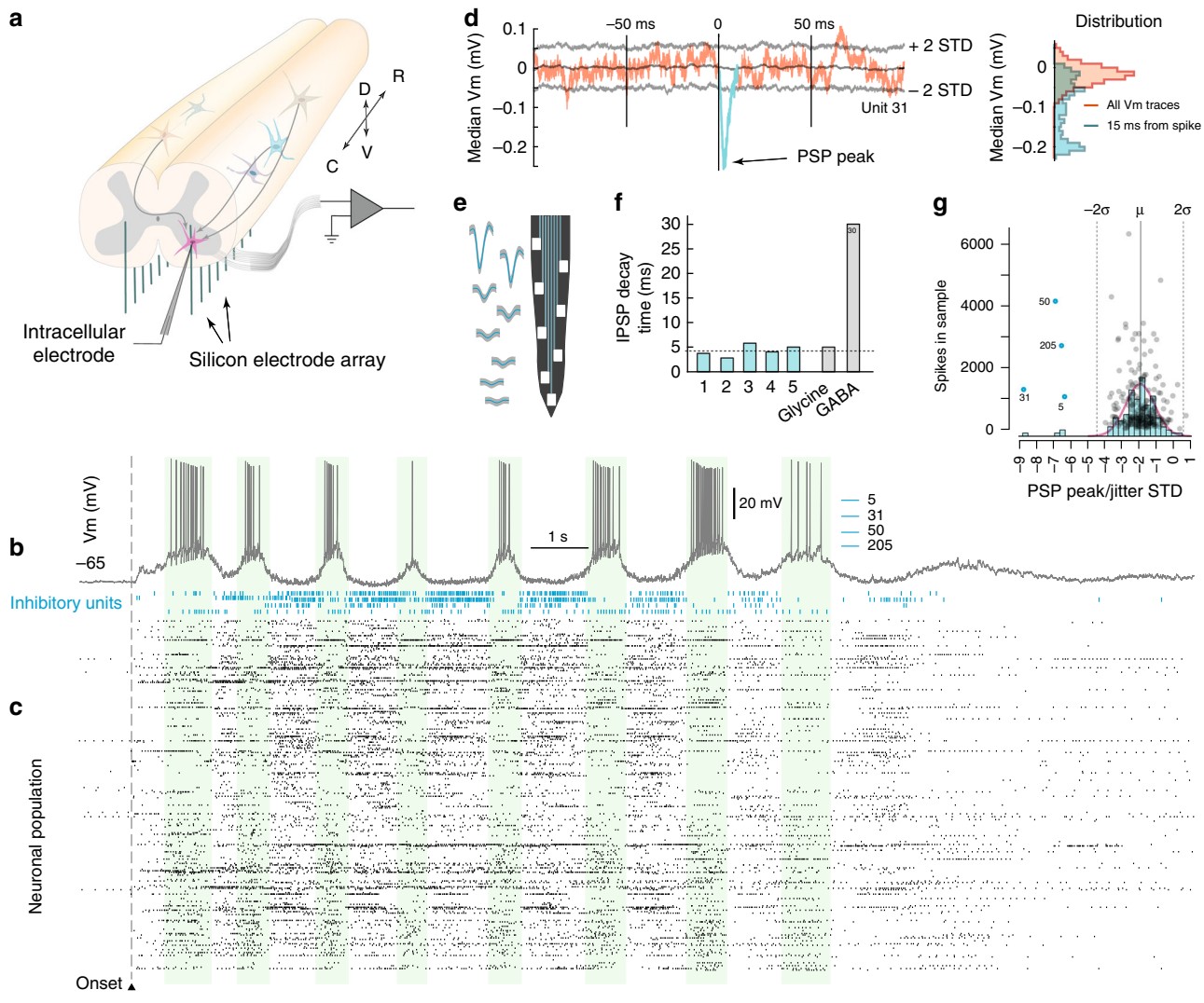

**Fig. 8** The lumbar connection probability is small. **a** Local inhibitory and excitatory connections are verified by combining intra–cellular recording ($V_m$) with extracellular multi–electrode arrays (inserted ventrally). **b** $V_m$ during motor activity with the nearby neuronal population (**c**, $n = 249$). Four inhibitory cells were identified (blue rasters). **d** Spike–triggered median $V_m$ indicates an inhibitory post-synaptic potential (IPSP) with a significant peak (cf. red and blue traces). Gray lines indicate ±2 standard deviations of a jittered distribution. **e** Waveforms of an inhibitory cell **d** on shank electrodes. **f** IPSP decay time constants for the significant 5 units (green) indicate glycinergic rather than GABAergic inhibition (gray). **g** PSP peaks normalized by the jitter distribution standard deviation scatter plotted against the number of spikes in the sample. Only 4 units (blue) were significant in this data set

suggests that active decorrelation could be part of the explanation in the absence of correlated input. The asynchronous state in cortical networks is achieved by a mechanism known as active decorrelation[41]. Active decorrelation relies on some sort of balance of excitation and inhibition in the network, potentially recurrent inhibition in the premotor network[39]. To test the conditions of such a scenario, we amended the model presented previously (Fig. 2). First, the feedforward network had low sparseness ($\rho = 0.5$), which gave correlations between receiver neurons of ~0.5 as before (Fig. 9a). Next, adding feedforward inhibition with the same sparseness ($\rho = 0.5$) to a local inhibitory population, which then connected to the receiver neurons ($\rho_{inh} = 0.9$) did not provide decorrelation of the correlated feedforward input (Fig. 9b). However, when adding recurrent inhibition within the inhibitory population itself, a substantial decrease in correlation was observed (cf. middle and bottom correlation matrices, Fig. 9c). Population histograms of these 3 scenarios illustrates the difference (Fig. 9d). Nevertheless, this strong decorrelation from recurrent inhibition was contingent on the sparseness of the inhibitory population. If the connectivity was

too dense (gray region, Fig. 9e), the correlations would be enhanced, whereas if the connectivity was sparse (red region), there was a decorrelation of the input. The sparseness used in (c), was $\rho = 0.9$, (circle). This is because dense divergent connections from the inhibitory population would actually enhance rather than suppress correlations. Another requirement for active decorrelation, is that the inhibitory population is actually active. Hence, the decorrelation is also activity–dependent. When the firing rate of the source population is low, there is little or no suppression of synchronous input (Fig. 9f).

If active decorrelation is the cause of the low pair-wise correlation we have observed, a time dependence of fast correlation should be visible in our data since the firing rate is waxing and waning during the rhythmic activity. To investigate this further, we simulated the correlation between pairs of receiver cells during oscillatory input drive (Fig. 10a). The pairwise correlation had a clear rhythmic dependence, due to the change in firing rate of the inhibitory population (Fig. 10b). In the off–cycles the activity of the inhibitory population was lower and therefore the suppression of the synchronized feedforward input

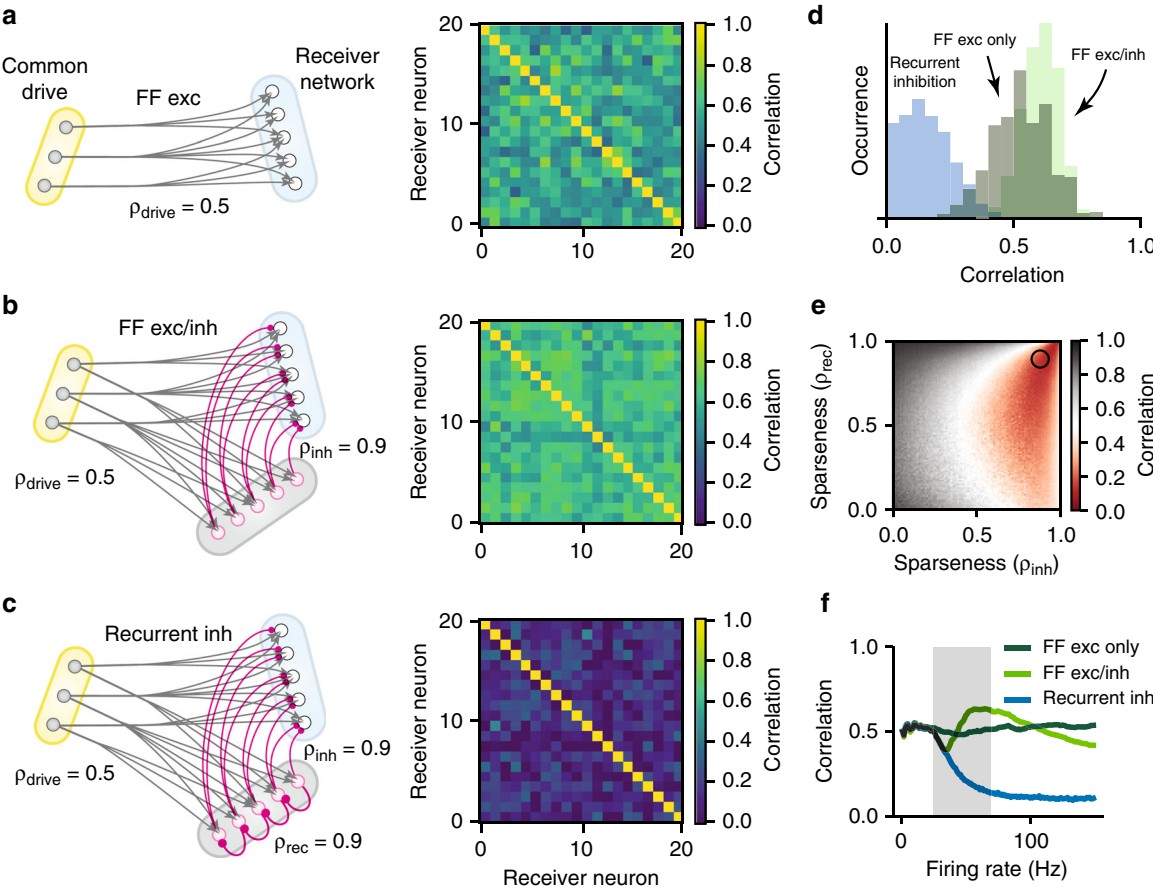

**Fig. 9** Active decorrelation by local inhibitory network requires sparse recurrent inhibition. Considering 3 network motifs: **a** Common drive network (yellow) with pure feedforward excitation (FF exc) with low sparseness ($\rho_{drive} = 0.5$) and pairwise correlation of ~0.5 (pairwise correlation matrix, right). **b** Including feedforward inhibition (inh) via a local inhibitory network (gray) has a similar correlation matrix, i.e. no decorrelation (right). **c** The inclusion of recurrent inhibition causes decorrelation in the receiver population (blue matrix, right). **d** Histograms of pairwise correlations for the 3 motifs, where the recurrent inhibition (blue) causes lower correlation. **e** Decorrelation (red) depends on connection sparseness, both from inhibitory to receiver cells ($\rho_{inh}$) and within the inhibitory population ($\rho_{rec}$). Circle indicates sparseness of **c**. **f** Decorrelation depends on the firing rate of the source neurons (gray region), especially for the recurrent topology (blue)

was absent, whereas during the on–cycles the input was decorrelated. The rhythmicity in the decorrelation was quantified by correlating the fast pairwise correlation (blue trace Fig. 10b) with the slow low–pass filtered $V_m$ (broken line Fig. 10a), which showed a negative correlation for the population compared with a shuffled distribution (cf. green vs. gray Fig. 10c). However, similar negative correlation between mean $V_m$ and the fast correlation was not seen in the experimental data. Pairs that were correlated on a slow timescale ($R > 0.9$) did not show a dependence between their fast timescale correlation and the low-pass filtered $V_m$ (in comparison with shuffled data) (Fig. 10d). Hence, we conclude that either active decorrelation by a local recurrent inhibitory population is not present in the spinal network, or the firing rate within this population is in the high end where there is no dependence (blue curve to the right of the gray region, Fig. 9f).

## Discussion

Although substantial progress has been made in describing spinal cell types and projection patterns[2,4,32,42–45] remarkably little is known about connectivity in motor circuits. Indeed, the size and extent of the neuronal population involved in generating motor activity are unknown, and essential features of graph theory, such convergence versus divergence, degree distributions[29], and

sparseness remain open issues. In this report, we address the network architecture of spinal CPGs from a dynamics perspective by employing intra and extracellular recordings from pairs of lumbar INs and MNs. We find a remarkable absence of correlation of input across all pairs, even for pairs that are strongly correlated on a slow timescale, and therefore belong to the same functional module. This paradoxical finding can be interpreted in two ways: (1) the common drive network is much larger than previously assumed, and the driving neurons are sparsely connected and convergent upon the receiving lumbar neurons; or (2) there is a pervasive *active decorrelation* cancelling an otherwise correlated drive from a smaller and denser network. Active decorrelation has been found among correlated sensory input to cortical networks[23–26]. It is worth noting that these two interpretations are not mutually exclusive. A network, which possesses both active decorrelation, as well as sparse connectivity, is indeed possible[41]. A circuit motif, which could participate in active decorrelation by combining feedforward excitation with inhibition, has been reported in the vertebrate spinal cord[46]. Nevertheless, the latter interpretation implies a radically different network structure, since such an asynchronous state is characterized by widespread recurrent inhibition and excitation[21,23], which is fundamentally dissimilar to the conventional feedforward scheme of spinal motor networks.

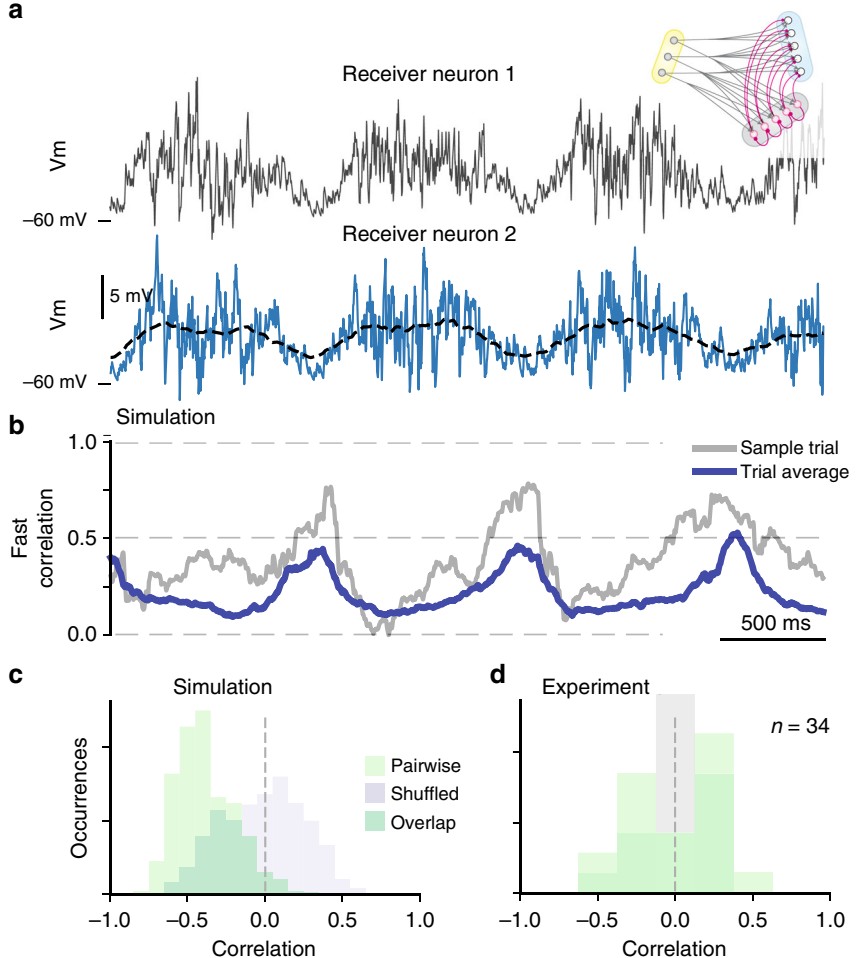

**Fig. 10** Active decorrelation is dependent on the firing rate. **a** Membrane potential ($V_m$) of two simulated sample neurons (gray and blue) for 3 cycles of motor behavior in a recurrent topology model (inset). The dashed line indicates low–pass filtered $V_m$ used for correlation analysis below. **b** Fast correlation between traces in **a** vs. time has a clear rhythm, i.e. dependence of firing rate. Sample pair (gray) and population average (blue). **c** Histogram of correlation between fast correlation and $V_m$ indicates anti–phase (negative correlation, green) compared with shuffled data (gray). **d** No anti–phase relation between fast correlation and $V_m$ was apparent in experimental measurement (green, $n = 17$ pairs, 34 correlations) i.e. not significantly different from shuffled distribution (gray, Kolmogorov Smirnov test)

Although most networks in the brain are sparse, likely due to their low wiring cost[29], such topology remains to be verified in spinal motor networks. Spinal CPGs have been suggested to rely either on unit oscillators[2,31], or a multilayered half–center model with centers, consisting of neurons with similar properties, connected in a feedforward manner[12,32] (Fig. 1). The internal connectivity in a module has been proposed to consist of glutamatergic neurons with recurrent connectivity to induce reverberate internal activity[47]. The lower modules are driven by a descending common network, which is responsible for the rhythm generation. This feedforward divergent wiring from a small neuronal population, which orchestrates a larger population, would result in a substantial overlap in synaptic input (Fig. 1b). Since we did not find the correlated activity that such overlap would cause, our data support either a sparse feedforward network with convergent connections or a recurrent network architecture with active decorrelation.

Although the purpose of a sparse network architecture is non-obvious, it may be relevant for controllability. The circuitry in the spinal cord produces overall motor activity, but supraspinal centers, especially the primary motor cortex and the brainstem, can exert a major influence on this activity. The pertinent question is what type of spinal circuit architecture best accommodates

the influence from supraspinal regions? Controllability is an active topic of investigation in graph theory, and it is defined as the ability to drive the network from any intrinsic state to any other state. It was suggested that sparse architectures are difficult to control compared with dense architectures[48]. Here, hubs[29] and modules may serve as intermediaries for supraspinal input. Finding modules in sparse networks is generally difficult[49], but an obvious location to inspect is where the corticospinal projections terminate. The issue of how such few supraspinal fibers can control motor behavior remains to be investigated and understood from a network perspective.

A surprising absence of correlation across a neuronal population, which is known to receive shared input, has been observed in cortical networks[23–26]. This enigmatic observation has motivated theoretical studies in active decorrelation since it was suspected to be the source. If inhibitory feedforward and feedback loops are present in networks, as recent data suggest[39,50], they will participate in the decorrelation of an otherwise correlated feedward input[30,41]. The recurrent inhibition[21], which is found in theoretical sparse networks that have balanced excitation and inhibition, can explain a low cross–correlation, which would otherwise be large[41]. It was proposed that recurrent inhibition is likely responsible for the active decorrelation, and the purpose of

such decorrelation is primarily to reduce noise from correlated excitatory drive.[41] A similar mechanism could play a role in spinal networks, where a clear advantage exists of reducing the correlated noise to ensure stable and smooth movements. This introduces a novel purpose of inhibitory interneurons in spinal circuits, which has not previously been considered. Spinal inhibition has traditionally been associated with sculpting of the motor rhythm via reciprocal connections[2,12] or modulation of gain during motor control[20]. Nevertheless, since active decorrelation occurs in networks with a balance between excitation and inhibition[23,24], and such concurrent E/I activity has been observed in spinal motor networks under certain circumstances[39,51–53], it is quite possible that part of the low correlation could be explained by this mechanism. In our simulations of active decorrelation (Figs. 9 and 10) we found that this scenario is likely only possible if the recurrent inhibitory population is itself sparse and substantially active. This inherent network sparseness could also explain the large number of interneurons, which is eightfold more numerous than MNs[54]. Such a new perspective on the purpose of inhibition in spinal networks remains to be substantiated in future experiments.

The population of neurons investigated in the current study was classified according to their electrical activity, which is agnostic towards their genetic identity. Although, spinal interneurons are often categorized according to their cellular linage, it is unclear if such categorization would have a simple relationship with the electrical activity. There is no indication of exclusive connectivity between interneuron subtypes and the various motor pools. Multiple clades of interneurons can innervate a single motor pool, and the same clade can innervate multiple motor pools[55]. Further, it is uncertain if cells of same subtype should have a preference of being interconnected, rather than being connected to cells of other origin. A recent investigation using intracellular recordings from pairs of interneurons with the *shox2* transcription factor demonstrated the internal connectivity among those interneurons to be sparse[56]. These interneurons together with those expressing the *Hb9* transcription factors have been suggested to be responsible for rhythm generation[57–59] and could represent the source neurons in the feedforward network presented here. Nevertheless, the motor circuitry is likely to be made up of a heterogeneous population with various electrical activities interconnected in a complex manner, which remains to be unraveled[39].

## Methods

**Definition of sparse networks**. The *density* (or *connectance*), $\xi$, of a network is often defined as the number of connections, $C$, over the maximum possible number of connections $C_{max}$[28]. In our model (Fig. 1b, c) the number of neurons is $n_1$ in the source network (yellow region) and $n_2$ for the receiving network (blue region). The maximal number of connections is then $C_{max} = n_1 \cdot n_2$. We assume that the number of incoming connections to the receiving neurons, i.e. the in–degree $k$, is constant regardless of $n_1$ and $n_2$. The total number of connections is therefore $C = k \cdot n_2$ and $\xi$ is independent on the number of receiving neurons, $n_2$:

$$\xi = \frac{C}{C_{max}} = \frac{k}{n_1}$$

The formal definition of a *dense* network according to graph theory is a network where $\xi$ approaches a constant (>0) as the network size increases $n_1 \rightarrow \infty$. A *sparse* network is a network where $\xi \rightarrow 0$ as $n_1 \rightarrow \infty$[28]. In biological networks the size of a network cannot approach infinity or even be changed. Therefore the definitions of sparse and dense networks are less helpful. A practical definition of a sparse network we therefore suggest a network where the density $\xi$ is very low. Further, we define sparseness ($\rho$) as $\rho = 1 - \xi = 1 - k/n_1$, i.e. the inverse of the density. Other definitions of sparse connectivity is a low probability (~10%) of finding a connection between neurons[60].

**Model and simulation**. All simulations were done using the NEST simulator (http://www.nestsimulator.org) ver. 2.10[61]. The code for the simulations are available either on the lab web site (www.berg-lab.net) or from the corresponding author upon reasonable request. The model consisted of a population of

motorneurons implemented as leaky integrate-and-fire cell model with conductance based synapses, for which, spiking was disabled. The parameters of the cell model was set to have a membrane capacity of 250 pF, leak conductance of 16.67 nS and resting potential of −60 mV. Synapses were modeled as exponentially decaying conductances with a timescale of 1 ms and reversal potential 0 mV or −80 mV for excitatory and inhibitory synapses, respectively. Three different input scenarios were considered: (i) The motorneurons population received only feed-forward excitatory input from an uncorrelated presynaptic population modeled as an inhomogeneous Poisson process (Fig. 2), (ii) An additional population of inhibitory interneurons (identical to the motorneurons but with a spike threshold at −50 mV) provided feedforward inhibitory input, or (iii) same as (ii) but the inhibitory population was recurrently connected to achieve active decorrelation (Figs. 9 and 10). Parameters of the model differed between (Fig. 2) and (Fig. 9 and 10):

Figure 2: Peak conductances of excitatory synapses were set to 10 nS. The population of motorneurons consisted of 30 cells. Time–varying input the presynaptic input population was modeled using an inhomogeneous Poisson process with a mean firing rate of 12 spikes/s modulated with a 1 Hz sinusoidal oscillation with 10 spikes/s amplitude. The size of the presynaptic population was varied (in separate simulations) between 10 and 500.

Figures 9 and 10: Peak conductances of synapses were set to 5 nS and 15 nS for excitatory and inhibitory synapses, respectively, to create an approximate balance between excitatory and inhibitory input currents at resting membrane potential. The receiving population of motorneurons and the inhibitory interneuron population each consisted of 100 cells. Time–varying input the presynaptic input population was modeled using an inhomogeneous Poisson process with a mean firing rate of 80 spikes/s modulated with a 1 Hz sinusoidal oscillation with 70 spikes/s amplitude.

**Data analysis**. All data analysis was performed in custom designed procedures either in Matlab (Mathworks, R2014b) or Python (www.python.org). Spike sorting was performed using Spyking Circus[62] (Fig. 6) and KlustaKwik[63] (Figs. 7 and 8). The intracellular membrane potential recordings were digitally filtered using a 3–pole butterworth filter in both directions to cancel phase distortion using the 'filtfilt' function in Matlab. The fast activity was high pass filtered with cut off 5 Hz after removing any potential action potentials. The slow activity was band–pass filtered from 0.2–5 Hz. These signals were then cross–correlated in pairwise fashion using Pearson correlation:

$$R_{xy} = \frac{\sum_{i=1}^{n}(x_i - \bar{x})(y_i - \bar{y})}{\sqrt{\sum_{i=1}^{n}(x_i - \bar{x})}\sqrt{\sum_{i=1}^{n}(y_i - \bar{y})}} \tag{1}$$

where $x_i$ and $y_i$ are the two arrays of observations to be compared, $n$ is the number of observations, and $\bar{x} = \sum_{i=1}^{n}(x_i)/n$, is the sample mean. The confidence limits were calculated by comparing the correlation coefficients to those achieved by shifting one trace with a delay, which is randomly selected, i.e. the shuffled data. The 95% confidence limit was ±1.96σ from the mean for the shuffled distribution. Phase between the rhythmic activity of two neurons recorded intracellularly was calculated as the location of the peak in the cross–correlation function of the low–pass filtered $V_m$ trace shift in time. A shuffled correlation was used for comparison, in which any causal correlation was eliminated by randomly shifting one trace while leaving the other trace intact. In this way, we could establish the correlation expected purely by chance. The shuffled recording was constructed by shifting the trace in time, in a region that had similar statistics of synaptic intensity. The shift was chosen randomly from trial to trial between 500 ms to 2 s.

**Polar histograms of the spike triggered phase**. The nerve activity was estimated by convolving the nerve signal with a Gaussian kernel ($\sigma = 150\ ms$) similar to the estimates of spike rates. The instantaneous phase of the nerve activity was calculated from the analytic signal using the Hilbert transform. Before applying the transform the nerve activity was high pass filtered at 0.1 Hz. We only calculated the spike triggered phase on nerve signals when there was a ongoing activity (as seen in Fig. 6).

**Pairwise correlation in spike rates: slow and fast**. Spike rates were convolved with a broad and and narrow Gaussian kernels[64],

$$k(t) = \frac{1}{\sqrt{2\pi}\sigma}\exp\left(-\frac{t^2}{2\sigma^2}\right) \tag{2}$$

where $\sigma = 5$ ms and $\sigma = 150$ms, respectively, to capture the fast and the slow activity, respectively (Fig. 7). The spike rates were further high–pass filtered with a 3-pole Butterworth filter using a zero–phase filter ('filtfilt.m') function in Matlab, with a cut–off frequency of 0.3 Hz for the slow activity and 10 Hz for the fast activity. We wanted to only consider rhythmically active neurons in the analysis, since these are related to the motor activity and easy to group functionally. We test if units are rhythmically active in relation to the motor program by applying Rayleighs test for circular uniformity. The unit that did not have rhythmic activity was excluded using the Rayleigh test of circular statistics[65]. The test statistic was $z = NR^2$, where $R$ is the length of the average phase vector in polar coordinates, and $N$ is the number of spikes for a given unit. The test statistic was Rayleigh's test for

circular uniformity[66,67] with a significance level of 5%. The p–value was estimated as

$$p = exp\left[\sqrt{(1 + 4N + 4(N^2 - R_N^2))} - (1 + 2N)\right] \quad (3)$$

where $R_N = R \cdot N$. Using this level, $N = 1701$ units were excluded out of the a total of 5791 recorded units over 3 animals (Fig. 7). For these significantly rhythmic units the cross–correlation was calculated for all pairs, both on slow and on fast timescales.

**Local connectivity.** The combination of the intracellular recording from a single cell and the simultaneous recordings from hundreds of neurons recorded extra-cellularly can be extracted from previous reports[19,27,52], but are describe briefly here. Three animals were implanted each with three 64–channel probes (Berg64–probes, Neuronexus inc.) in D8, D9 and D10. These are lumbar segments in the turtle corresponding to L2-L5 in mammals[54]. In addition, an intracellular sharp electrodes was inserted from the ventral side and ipsilaterally to the probe in D10. Spike sorting was performed in open source software (Klustakwik-suite: SpikeDetekt, KlustaKwik v.3.0 and KlustaViewa[63]). Spike–triggered average (STA) and spike–triggered median (STM) membrane potential ($V_m$) of an intracellular recorded neuron was calculated for all simultaneous extracellular recorded neu-rons, typically 200–300 neurons. In order to minimize the impact of large excur-sions in $V_m$ due to the occurrence of action potentials, the median of $V_m$ was used and z-scored, i.e.

$$z_{V_m} = \frac{V_m - \mu_{V_m}}{\sigma_{V_m}} \quad (4)$$

The z-scoring compensates for the different degree of fluctuations across recorded neurons. In order to set an unbiased connection-threshold we chose STM over STA, as STM allowed us to consider spike triggered traces with spike in them, as the median value is less affected by the skewness of the $V_m$-distribution, introduced by spikes in the $V_m$-trace. Average traces without spikes in the spike triggered window, were observed to be visually indistinguishable from the median traces. To test whether the spike–triggered synaptic potentials where exceeding chance level, we compared the temporal structure in $z_{V_m}$ with a surrogate data set, where the spike times had been jittered. The location of the spike in time were locally jittered to abolish temporal structure using the interval jitter method[68] of size 100 ms. The same analysis was then performed to establish a distribution of $z_{V_m}$ when there is no causal structure.

**Experimental preparation.** The data that support the findings of this study are available either on the lab web site (www.berg-lab.net) or from the corresponding author upon reasonable request. 80 adult red-eared turtles (*Trachemys scripta elegans*) of both sexes were used in this study. Animal was placed on crushed ice for 2 h. to ensure hypothermic anesthesia, then killed by decapitation and blood substituted by perfusion with a Ringer solution containing (in mM): 120 NaCl; 5 KCl; 15 NaHCO$_3$; 2MgCl$_2$; 3CaCl$_2$; and 20 glucose, saturated with 98% O$_2$ and 2% CO$_2$ to obtain pH 7.6. The carapace containing the D4-S2 spinal cord segments (corresponding to the cervical to lumbar regions) was isolated by transverse cuts[20,27,51] and the cord was perfused with Ringer's solution through the *vertebral foramen*, via a steel tube and gasket pressing against the D4 vertebra. The surgical procedures comply with Danish legislation and were approved by the controlling body under the Ministry of Justice.

**Activation of motor program.** To reproducibly activate the scratching motor pattern, a linear actuator was applied to provide mechanical touch on the skin around the legs meeting the carapace. The somatic touch was controlled by a function generator (TT2000, Thurlby Thandar instrument, UK) and consisted of a ten-second long sinusoidal movement (1 Hz). The touch was applied on the border of the carapace marginal shields M9-M10 and the soft tissue surrounding the hindlimb, which is the receptive field for inducing pocket scratching motor pattern[69].

**Electrophysiology.** Each scratch episode lasted approximately 20 s. A new trial was initiated after a 5 min rest. Electroneurogram recordings (ENG) were performed with suction electrodes of the hip flexor nerve and dD8 at the level of D9-D10 vertebra. The ENGs were recorded with a differential amplifier Iso-DAM8. The bandwidth was 300 Hz–1 kHz. The transverse cut was performed at the caudal end of D10 of the spinal cord in order to get access to the motor– and inter–neurons[27]. Pairwise intracellular recordings were performed using sharp electrodes (≈40 $M\Omega$). Each neuronal pair was recorded for at least one trial, and typically 3–4 trials. Given the length of each scratch episode (20 s), a single trial is enough to estimate the pairwise correlation in $V_m$. The electrodes were filled with a mixture of 0.9 M potassium acetate and 0.1 M KCl. In most experiments the electrodes also con-tained 4% W/V biocytin to leave a stain in the cell for post hoc histology. All experiments were conducted in current-clamp mode with a Multiclamp 700B amplifier (Molecular devices, Union City, CA). Data were sampled at 10 kHz with a 16-bit analog-to-digital converter, controlled and displayed with Clampex software. Glass pipettes were pulled on a P-1000 (Sutter instruments, USA). Motoneurons

were accessed from the surface at a typical depth of 50–300 $\mu m$ using motorized micromanipulators.

**Multi–electrode recordings.** Extracellular multi–electrode recordings were per-formed in parallel at 40 KHz using a 256–channel multiplexed Amplipex amplifier (KJE-1001, Amplipex). Up to four 64-channel silicon probes were inserted in the spinal cord from ventral side in incisions in parallel to the spinal cord. We used the 64-channel probes (Berg64–probe from NeuroNexus Inc., Ann Arbor, MI, USA) with 8 shanks, and 8 recording sites on each shank arranged in a staggered con-figuration with 30 $\mu m$ vertical distance (Fig. 6a). The shanks are distanced 200 $\mu m$ apart. Recordings were performed at depths in the range of 400–1000 $\mu m$ inserted from the ventral side of the cord.

**Identification of motoneurons.** Motoneurons were mainly identified by their location in the ventral horn, size (via $R_m$), size of action potentials and spiking relation with nerve activity. A subset was filled with biocytin for histological processing. The tissue containing the motoneuron was carefully removed and left in phosphate buffered saline (PBS) with 4% paraformaldehyde for 24–48 h. The tissue was then rinsed with and stored in PBS. The tissue section was mounted in an agar mount and sliced into several 100 $\mu m$ slices using a microtome (Leica, VT1000 S). The slices were incubated for 3–4 hr at room temperature with Cyanine-3-conjugated (Cy3) to streptavidin (1:500 or 1:250 Jackson ImmunoR-esearch labs, Inc) in blocking buffer (PBS with 5% donkey serum and 0.3% Triton X-100). The slices were washed with PBS and incubated overnight at 4 ℃ with primary choline acetyltransferase (ChAT) antibodies goat anti-ChAT antibodies (1:500, AB144P, Millipore, USA) diluted in blocking buffer. The slice was washed three times with PBS and incubated for 1 hr at room temperature with the sec-ondary antibody Alexa488 conjugated to donkey anti-goat antibodies (1:1000 Jackson) diluted in blocking buffer. After three washes with PBS, the slice was mounted and coverslipped using ProLong Gold antifade reagent (Invitrogen Molecular Probes, USA) and cured overnight at room temperature before micro-scopy. Micrographs were produced using a confocal microscope, Zeiss LSM 700 with diode lasers, on a Zeiss Axiolmager M2 using a 20 × /0.8 Apochromat objective (Zeiss). The fluorophores were excited/detected at: Cy3 at 555 nm/ 559–700 nm, Alexa488 at 488 nm/405–544 nm, and DAPI at 405/420–700 nm. The pinhole was 35 $\mu m$ resulting in an optical section of 2 $\mu m$. For all the channels a mosaic of 5 × 6 was made. During the z-stack of Cy3 fluorescence 15 optical slices with slight overlap gave a total optical section of 28 $\mu m$. A maximum-intensity-projection of the Cy3 z-stack was done and superimposed on the DAPI and Alexa488 image. The DAPI and Alexa488 image was taken in the middle of the Cy3 z-stack. Images were handled with ZEN 2011 software (Zeiss) in the LSM and 8-bit TIFF format.

**Reporting Summary.** Further information on research design is available in the Nature Research Reporting Summary linked to this article.

## Data availability

The data that support the findings of this study are available from the corresponding author upon reasonable request.

## Code availabilty

The code that analyzed the neural data and performed the network simulations of this study are available from the corresponding author upon reasonable request.

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

## Acknowledgements

Thanks to P.A. Kirkwood and P. Roland for comments on an earlier version of the manuscript. Funded by The Independent Research Fund, Denmark (R.W.B., M.V., P.C.P. and M.R.) and Mobilex with European Union co–fund program (H.L.). The work is part of the Dynamical Systems Interdisciplinary Network, University of Copenhagen.

## Author contributions

Conceptualization, M.R, H.L. and R.W.B.; Methodology, Investigation, Software, Visualization, Formal Analysis, R.W.B., M.R., A.W., M.V., P.C.P., H.L.; Writing–Original Draft, Funding Acquisition, Supervision, R.W.B.

## Additional information

**Competing interests:** The authors declare no competing interests.

