## [Peer Review File · Nature Communications]

Reviewers' Comments:

Reviewer #1:

Remarks to the Author:

This manuscript tackles the broad question of how the spinal network is organised, and in particular the organisation of the connections from a putative pattern generator to the spinal inter- and motor-neurons. The connections from this afferent generator are inferred using pairwise intracellular recordings in the spinal cord of turtles during invoked rhythmic scratching. The absence of "fast" correlations between membrane potentials is interpreted as implying a sparse, divergent set of connections from the generator to the receiving MNs and INs; or implying active decorrelation of inputs by a local circuit.

There are some interesting results here. The total decoupling of slow and fast correlations of the membrane potential is a striking result (Fig 4-5), with a nice control using pair-wise recordings of the same neuron. The inference of the sparseness of local connectivity (Fig 8) is also interesting. And the use of a model to guide interpretation (Fig 2) is good to see.

That said, in its present form, the study and manuscript lack a little coherence. The opening question is framed as understanding the connections from a pattern generator to a "receiver" population, and this is how the model is constructed. However, the interpretation of the results is not made in this framework (e.g. it is often unclear whether the authors are referring to the generator, the receiver, or the entire network when discussing their results - see e.g. the final line of the abstract); and some of the results (eg Fig 8) are unrelated to the framework.

The biggest issue with coherence is that connections exist between "receiver" neurons (here spinal MNs and INs), and so the interpretation of the experimental results in the framing model is not logically consistent. After all, it could turn out that the afferent population is small and connections to the "receiver" are dense, but are strongly decorrelated by the local "receiver" network. Indeed, the authors turn to active decorrelation in the Discussion. But solutions to this logical inconsistency are already in the study: Figure 8 shows the deep sparseness of the receiver population, suggesting that the local receiver network cannot strongly decorrelate its own inputs.

Thus I'd suggest the following:

(1) Show in the model the effect of recurrent connectivity within the receiver population on the correlation of membrane potentials.

Given the data from Figure 8, and estimates of MN:IN ratio and conductances of the respective connections, repeat the simulations of Fig 2 in a recurrently connected receiver network, and determine their effect on the relationship between the density of input connections and the correlation of membrane potentials.

(2) Test robustness of this effect (if any) against varying strengths of I and E conductances

(3) Then further interpret results from Fig 4 in the context of these results

(4) Update manuscript: keep the initial framing (Figs 1->4); then turn to the problem of the connected receiver network, asking: does it change our interpretation? (Fig 8 becomes Fig 5, then

new results thereafter). Current Figs 6 and 7 are interesting, but add little to the main result (see below), so could easily be Supplementary Information

Minor comments:

(1) Figure 2:

- add note in legend on how the inputs are generated (i.e. why they have a rhythmic component)
- panels d and e have "predicted" lines. No mention in main text or in Methods of the equations derived to predict these relationships
- Define "fast" correlation in the legend

(2) Fast vs slow correlations: the analysis of intracellular data in Figure 4 separates the Vm correlations into fast and slow components; the analysis of Fig 2 mentions only fast correlations in the legend. To aid interpretation, compute fast and slow correlations for Fig 2: how do the slow correlations scale with sparseness?

(3) Fig 3: if pairwise distances between electrodes are known (I appreciate they may not be), it would be interesting here to bolster the claimed lack of physical relationship with "functional" modules by seeing the distribution of pair-wise phase vs distance

(4) Fig 4 and pg 5: the text mentions no clear relationship between the time-series of "fast" correlations and the motor rhythm (the "slow" signal). The authors could check this explicitly by correlating the time-series to a reference nerve signal as they do in Fig 6. Again, an optional suggestion.

(5) Figs 6-7: the distribution of spiking phases (Fig 6) is interesting, but does not seem to contribute evidence for network sparseness etc; that millisecond precise spike-to-spike correlations are absent (Fig 7) is worth establishing, but hardly surprising (after all, if Vm has no coupling, spikes cannot either). These could be placed in Supplemental Info.

(6) All experimental data figures (3->8): it is rarely clear how many preparations contributed data, and no figure or text mentions how many repeats of the evoked scratching were used to generate the measured results.

(7) Fig 8: 5 inhibitory units are mentioned in the text, legend, and shown in panel f; but only 4 are highlighted in panel c.

Text issues and typos:

Abstract: "excitatory conductivity" -> "connectivity"

pg 1: "underlying the activity": what "activity" is being referred to?

pg 1: "size of such a network is ... responsible..": unclear what this means

pg 2: "different from the previously assumed." The sentence lacks a noun - from the assumed what?

pg 6: "the network must be convergent..." presumably "divergent" was meant here?

Fig 6 legend: panel (e) not referenced

Methods:

- Both Spyking Circus and the KlustaKwik suite were used for spike-sorting: which data were used with which approach?

- line after Equation 2: "fast" and "slow" are switched

Reviewer #2:

Remarks to the Author:

Radosevic et al attempt to tackle the interesting question of how a neuronal circuit is wired up, in either a sparse, convergent or dense convergent manner. The dual intracellular recordings and the combined multi-electrode array and intracellular recordings are technically demanding. The data analyses are extensive and equally challenging. While impressed with the data quality and analyses, the experimental design and interpretation of data are not very suitable for supporting the conclusions they have made.

The presumption of this type of connectivity analysis (Fig.1) is that when you are recording the postsynaptic neurons (Fig.5), they belong to the same functional group (module in this paper), and when you are recording from the presynaptic neuron (Fig.6-8), they are also from the same functional group, i.e. they form synapses on the neurons with the same physiological roles. Unfortunately, the spinal circuit is not organised in clear nuclei or layers per se, making selective recording from the same functional group of neurons using conventional dual intracellular recordings and multi-electrode array technically unachievable. In theory, the more functionally diverse the recorded neurons are, the less likely a correlation will be found.

To partially remedy this difficulty in meeting such presumptions in the chosen preparation, one could make many recordings and use post-experimental screening to identify possible neurons belonging to the same functional pool. This can be justified to some extent because rhythmic motor activity is mostly phasic. It is likely that neurons recorded with highly similar phase lags during scratching (i.e. recordings with high slow scale correlations) may belong to the same functional group. It also helps if the recordings are made from exactly the same location across preparations. This means the vast majority of existing recordings (Fig.5-8) need to be ditched for the analyses and a lot more recordings are going to be needed. Take Fig.8 as just one example, the postsynaptic cell is the same neuron so there is no violation of the postsynaptic presumption. However, the random recording of many presynaptic neurons using the electrode array may sample neurons from many different presynaptic pools. This will inevitably lead to very low connectivity probability for inhibitory neurons.

Why should the correlation values for slow and fast scale signals be locked in a linear manner (Fig.5a, 7c)? Such relation may be exponential, for example. Therefore it will be interesting to analyse the points where correlation of slow scale signal is >0.8 and carry out some data transformation to re-examine the relation, rather than using a unity line which covers from -1 to 1.

Fig.1 State: grey symbols are local premotor neurons and white ones are motoneurons (MN)

Last paragraph on p2, citation of Fig.1b, Fig.1c is incorrect.

Fig.2, $n=20$ cells but the histogram suggests more cells.

Fig.3d, label IN, MN instead of neuron3, neuron 4.

Fig.4, explain "shuffled data correlation".

Reviewer #3:

Remarks to the Author:

This interesting paper considers the broad question of correlations in spinal motor neurons and interneurons during rhythmic activity and its implications for network connectivity and mechanisms. The preparation used is that of the turtle spinal cord, during hindlimb scratching, which offers a number of unique advantages, as a result of which the results reported here are the first of their kind in many respects. The work is rigorous and well-presented.

Before getting to detailed comments/questions, I want to mention that as someone not active in the subfield of motor circuits, some of my comments are possibly addressed by short remarks and appropriate citations, that would also serve the non-aficionado reader.

Comments/questions:

1. The authors state that their results are consistent either with the possibility of sparse convergent connectivity and/or a mechanism that performs active decorrelation. The title however only reflects the former possibility. The authors should consider either hedging the title a bit to accommodate the latter possibility or perhaps have a more general title that does not mention either. In short, my concern is that if future work in fact shows evidence for the active decorrelation, the title would not age well.
2. The experimental preparation requires decapitation and removal of muscles that limit proprioceptive feedback. It would be useful to see a short discussion of how experimental results so obtained might differ from that of the same circuits in a fully intact animal.
3. The model, as I understand it, considers the case where the source neurons are themselves uncorrelated. It would be useful to see analysis (described possibly in Supplementary) of the case where the source neurons are also correlated, and how that affects the plots. of Fig 2.
4. Apropos Fig 3(e), are all neuron pairs oscillating in the same frequency? Were pairs of non-commensurate frequency found? I don't recall seeing this being addressed.
5. To what extent is the lack of fast-timescale correlations due to physiological variability between cells? As I understand it, the present model has cells with identical parameters. Some modeling to address this would be insightful. For example, from the data, one question could be: Is the ratio of spike counts across peaks of the slow timescale between neuron pairs somewhat preserved?
6. Is there a reason the distribution of slow time scale correlations is markedly different between Fig 7d and Fig 5b?
7. The video abstract is nicely done and provides a good, quick, first overview of the work. I would suggest retaining it.
8. I appreciate the authors starting with the model that forms the backdrop for the rest of the paper.

Minor comments:

1. Page 5, last paragraph: "separation" is misspelled as "seperation".

2. Fig 6 legend: Panel (e) hasn't been explicitly mentioned.

In closing, this is a strong paper with comprehensive data and interesting results that will spur further theory and experiment. I would recommend that the paper be accepted with a minor revision.

Reviewers' comments:

Comment to all reviewers:

We have performed a thorough revision of the manuscript and addressed all the points raised by the reviewers. In particular we have expanded the modelling to include the issue of active decorrelation, raised by reviewer 1. We have also included some references, which we missed in the previous round and which are relevant, in particular, the work by Song and colleagues, who have performed pairwise recordings in the spinal cord. (Song, J., Dahlberg, E. & El Manira, A. V2a interneuron diversity tailors spinal circuit organization to control the vigor of locomotor movements. *Nature Communications* 9, 3370 (2018). Song, J., Ampatzis, K., Bjornfors, E. R. & El Manira, A. Motor neurons control locomotor circuit function retrogradely via gap junctions. *Nature* 529, 399–402 (2016).)

Reviewer #1 (Remarks to the Author):

This manuscript tackles the broad question of how the spinal network is organised, and in particular the organisation of the connections from a putative pattern generator to the spinal inter- and motor-neurons. The connections from this afferent generator are inferred using pairwise intracellular recordings in the spinal cord of turtles during invoked rhythmic scratching. The absence of "fast" correlations between membrane potentials is interpreted as implying a sparse, divergent set of connections from the generator to the receiving MNs and INs; or implying active decorrelation of inputs by a local circuit.

There are some interesting results here. The total decoupling of slow and fast correlations of the membrane potential is a striking result (Fig 4-5), with a nice control using pair-wise recordings of the same neuron. The inference of the sparseness of local connectivity (Fig 8) is also interesting. And the use of a model to guide interpretation (Fig 2) is good to see.

Thank you for the constructive comments, and for taking the time to review our manuscript.

That said, in its present form, the study and manuscript lack a little coherence. The opening question is framed as understanding the connections from a pattern generator to a "receiver" population, and this is how the model is constructed. However, the interpretation of the results is not made in this framework (e.g. it is often unclear whether the authors are referring to the generator, the receiver, or the entire network when discussing their results - see e.g. the final line of the abstract); and some of the results (eg Fig 8) are unrelated to the framework.

This is a good point. Since the general consensus is that the generator/driving population is small, i.e. a common-drive network, the chance that we would be recording from a neuron belonging to this driving network is also small. Hence, we assume that the neurons that have not been confirmed to be motoneurons, are inter-neurons, that do not belong to the driving network. We have gone through the manuscript to make sure it is more clear what population we thinking about in the interpretation. We also make the assumption explicitly clear in the manuscript with the following sentences in the introduction:

“First, we utilize dual intracellular recordings to assess the strength of synaptic correlations, in particular for pairs belonging to the same module. The modular–association is based on two issues: 1) motor neuron pairs in close vicinity and with same slow phase 2) interneurons also in close vicinity that have same phase are assumed to belong to same module and receive common drive. This assumption is based on the consensus view that the common–drive network is small compared with the receiver network (hence the term ‘common’) and therefore the risk of randomly recording from one of the source–network neurons is equally small. Next, we use multi–electrode arrays to measure population activity to determine the pairwise spike–spike correlation as an alternative indicator for shared synaptic input, under same assumptions.”

You may argue that our data show that the drive network is not small, and therefore this assumption is undermined. Nevertheless, this would precisely reject the hypothesis of a small kernel responsible for feedforward driving the rest of the network, which is the main conclusion of the paper.

To avoid mixing the issues, we have now removed the following sentence from the abstract:

“The inhibitory connectivity was $< 1\%$, and the excitatory connectivity was even lower.”

The biggest issue with coherence is that connections exist between "receiver" neurons (here spinal MNs and INs), and so the interpretation of the experimental results in the framing model is not logically consistent. After all, it could turn out that the afferent population is small and connections to the "receiver" are dense, but are strongly decorrelated by the local "receiver" network. Indeed, the authors turn to active decorrelation in the Discussion. But solutions to this logical inconsistency are already in the study: Figure 8 shows the deep sparseness of the receiver population, suggesting that the local receiver network cannot strongly decorrelate its own inputs.

Yes - this is an interesting point, thank you for bringing that up. The low connectivity probability illustrated in Fig. 8, could suggest active decorrelation. We have elaborated on this point and added analyses, both modelling and experimental in the manuscript, see below.

Thus I'd suggest the following:

(1) Show in the model the effect of recurrent connectivity within the receiver population on the correlation of membrane potentials. Given the data from Figure 8, and estimates of MN:IN ratio and conductances of the respective connections, repeat the simulations of Fig 2 in a recurrently connected receiver network, and determine their effect on the relationship between the density of input connections and the correlation of membrane potentials.

We have now performed simulations of a model similar to Fig. 2, now augmented with local recurrent connectivity, that decorrelates a correlated input. This results in two new figures (fig. 9 and 10) and a result section. Basically, we find that an inhibitory feedforward network with recurrent connections, is indeed able to decorrelate an otherwise correlated input to motor neurons. However, we find two new issues: 1) the decorrelation is activity-dependent (if the inhibitory population don't spike, they cannot decorrelate anything), and therefore we should see a waxing and waning of the decorrelation going through the cycle. 2) If the inhibitory connectivity is dense, it actually enhances synchrony- thus suppression of correlation requires sparse connectivity within the inhibitory population. Regarding 1), we also reanalyzed the experimental

data, to look for such a waxing and waning and we did not find it (Fig 10d). This could be explained by the inhibitory population being in a high-firing regime. We have adapted the text and the discussion accordingly.

(2) Test robustness of this effect (if any) against varying strengths of I and E conductances

We tested the robustness of the decorrelation by changing the connectivity rather than the E and I conductances themselves. We like to consider this additional element for a future study, where we have more space to treat it properly (we already use 10 figures).

(3) Then further interpret results from Fig 4 in the context of these results

Yes- We have updated the manuscript in the light of these new results, please see 1) above.

(4) Update manuscript: keep the initial framing (Figs 1->4); then turn to the problem of the connected receiver network, asking: does it change our interpretation? (Fig 8 becomes Fig 5, then new results thereafter). Current Figs 6 and 7 are interesting, but add little to the main result (see below), so could easily be Supplementary Information

Yes- We have updated the manuscript. However, we are a bit confused about taking out figure 5, which is our key result. Did you mean figure 6? We still think Figure 6 is important: First, it is a method figure that explains the experiment, how are the units acquired and sorted (polytrode sorting, with 3 sample units shown). Second, the figure shows sample spike dynamics in the cycles- a type of data that is very rare in the literature of spinal motor activity.

Regarding figure 8, this is a result based on the slow and fast pairwise correlation of spikes, which is complementary to the main result (figure 4-5). Since the journal allows 10 display items, and the figures 6 and 7 still serve an important purpose we suggest leaving them in rather than putting them in supplementary information. Hence, we add two figures (Figure 9 and 10, see below) in the end, with the new model with recurrent inhibition, that actively decorrelates the otherwise correlated input.

FIGURE 9:

CAPTION: “Active decorrelation by local inhibitory network requires sparse recurrent inhibition. Considering 3 network motifs: (a) Common drive network (yellow) with pure feedforward excitation with low sparseness ($\rho_{drive} = 0.5$) and pairwise correlation of ~ 0.5 (pairwise correlation matrix, right). (b) Including feedforward inhibition via a local inhibitory network (gray) has similar correlation matrix, i.e. no decorrelation (right). (c) Inclusion of recurrent inhibition causes decorrelation in the receiver population (blue matrix, right). (d) Histograms of pairwise correlations for the 3 motifs, where the recurrent inhibition (blue) causes lower correlation. (e) Decorrelation (red) depends on connection sparseness, both from inhibitory to receiver cells (ρ_{inh}) and within the inhibitory population (ρ_{rec}). Circle indicates sparseness of c. (f) Decorrelation depends on firing rate of the source neurons (gray region), especially for the recurrent topology (blue). “

FIGURE 10:

CAPTION: Active decorrelation is dependent on firing rate. (a) Membrane potential (V_m) of two simulated sample neurons (gray and blue) for 3 cycles of motor behavior in a recurrent topology model (inset). Dashed line indicates low-pass filtered V_m used for correlation analysis below. (b) Fast correlation between traces in (a) vs. time has a clear rhythm, i.e. dependence of firing rate. Sample pair (gray) and population average (blue). (c). Histogram of correlation between fast correlation and V_m indicates anti-phase (negative correlation, green) compared with shuffled data (gray). (d) No anti-phase relation between fast correlation and V_m was apparent in experimental measurement (green, $n = 17$ pairs, 34 correlations) i.e. not significantly different from shuffled distribution (gray, KS2 test).

Minor comments:

(1) Figure 2:

- add note in legend on how the inputs are generated (i.e. why they have a rhythmic component)

We have modified the following sentence in the caption:

“Oscillatory spiking activity of the motor rhythm was imitated as an inhomogeneous poisson process with sinusoidal rate in a small common

drive model network..."

- panels d and e have "predicted" lines. No mention in main text or in Methods of the equations derived to predict these relationships

Yes. We now added some text in the caption:

"(d) Mean fast correlation (* in c) is inversely linked to sparseness (circles). The expected correlation (orange, 'predicted') is the fraction of common connections over the total number of synaptic input. (e) Sparseness (gray) increases and fast correlation ('predicted', orange, 'simulated' circles, as in d) decreases with network size."

- Define "fast" correlation in the legend

Yes. We have added a note so the following sentence reads:

"...neurons with substantial correlation on both fast and slow timescales (below/above 400ms-timescale)."

(2) Fast vs slow correlations: the analysis of intracellular data in Figure 4 separates the Vm correlations into fast and slow components; the analysis of Fig 2 mentions only fast correlations in the legend. To aid interpretation, compute fast and slow correlations for Fig 2: how do the slow correlations scale with sparseness?

Yes, we have now calculated the slow correlation as well. We have added the histogram for the slow correlation (a bar at 1) and added the following text in the caption:

"The slow correlation remains 1 in both topologies (cyan)."

We thought about adding a constant line = 1 in the figure to illustrate the dependence of slow correlation with sparseness, but decided it would confuse more than help.

(3) Fig 3: if pairwise distances between electrodes are known (I appreciate they may not be), it would be interesting here to bolster the claimed lack of physical relationship with "functional" modules by seeing the distribution of pair-wise phase vs distance

Yes, that would be interesting. Unfortunately, we do not have the exact distance between somata, only that it is less than 300 microns. We hope to be able to do this in a future study.

(4) Fig 4 and pg 5: the text mentions no clear relationship between the time-series of "fast" correlations and the motor rhythm (the "slow" signal). The authors could check this explicitly by correlating the time-series to a reference nerve signal as they do in Fig 6. Again, an optional suggestion.

We have now done this on all the data, since it is important for the interpretation of the new model. We used the slow Vm rather than the nerve, so avoid the confounding factor of a potential phase lag of the nerve activity. There was no significant difference between the distribution of correlation between Vm(slow) vs the correlation of the fast activity in a shuffled surrogate data comparison. The processed data is shown as histograms in Figure 10d now, in the revised manuscript.

(5) Figs 6-7: the distribution of spiking phases (Fig 6) is interesting, but does not seem to contribute evidence for network sparseness etc; that millisecond precise spike-to-spike correlations are absent (Fig 7) is worth establishing, but hardly surprising (after all, if Vm has no coupling, spikes cannot either). These could be placed in Supplemental Info.

The assessment that the Vm has no coupling is based on pairwise recording of 66 pairs, which is a large number, but is nevertheless still a small number compared with the possible number of pairs of the motor network. The purpose of including the spiking data is to drastically increase the number of pairwise comparison, although in a different way than the intracellular data. This data serves as a complementary analysis to substantiate the finding. Figure 6 also helps the reader understand how the data was acquired. Keep in mind multielectrode-arrays inserted into the spinal cord and polytrode sorting have rarely been done before. We suggest leaving these figures in the manuscript, to help the reader and the integrity of the manuscript. The journal allows 10 display items, so space is not an issue in this context.

(6) All experimental data figures (3->8): it is rarely clear how many preparations contributed data, and no figure or text mentions how many repeats of the evoked scratching were used to generate the measured results.

We have added the following text in the methods section, which should help the clarity:

“Pairwise intracellular recordings were performed using sharp electrodes ($\approx 40\text{M}\Omega$). Each neuronal pair was recorded for at least one trial, and typically 3–4 trials. Given the length of each scratch episode (20 sec), a single trial is enough to estimate the pairwise correlation in V_m .”

(7) Fig 8: 5 inhibitory units are mentioned in the text, legend, and shown in panel f; but only 4 are highlighted in panel c.

Yes- this is because the fifth unit was identified in a different intracellular recording. We now specify this by changing the following text in the last paragraph before discussion:

“The significance of the synaptic connections was established by comparison of the IPSP–peak with that of a surrogate data, where a temporal structure had been abolished by random jitter of the spike–times (Fig. 8g). The inhibitory activity is shown in blue (Fig. 8c), ...”

is changed into:

“The significance of the synaptic connections was established by comparison of the IPSP–peak with that of a surrogate data, where a temporal structure had been abolished by random jitter of the spike–times. The PSP peak distribution for a data set containing 4 such inhibitory units is shown (Fig. 8g). The corresponding spike activity for the same data set containing the 4 inhibitory activity is shown in blue (Fig. 8c),... “

Text issues and typos:

Abstract: "excitatory conductivity" -> "connectivity"

Done!

pg 1: "underlying the activity": what "activity" is being referred to?

Referring to the generation of motor program. We have modified the sentence accordingly:

“network connectivity underlying the activity” -> “the connectivity of

the network responsible for generating the motor activity remains unknown.”

pg 1: "size of such a network is ... responsible..": unclear what this means

Yes it is unclear. We have changed the sentence:

“Although the size of such a network is known to be responsible for the respiratory drive center “ -> "Although the size of the respiratory motor network, i.e. the preBotzinger complex, is well-known, the size and wiring of other CPG networks are not well understood.

pg 2: "different from the previously assumed." The sentence lacks a noun - from the assumed what?

We have change the last part of the sentence accordingly:

"different from the previously assumed." -> “which is fundamentally different from the previously assumed role of inhibition in the spinal cord.”

pg 6: "the network must be convergent..." presumably "divergent" was meant here?

We dont see why you think it should be divergent? If you see the fig. 2e, the mean pairwise correlation (the orange line) declines toward zero as the network size increases. This is also the case when the sparseness increases, the mean correlation decreases. I guess we have not plotted the convergence as a function of correlation, but sparseness and convergence go hand in hand, if the target neurons has a fixed number of synaptic contacts.

Fig 6 legend: panel (e) not referenced

Panel (e) is now described in the caption:

“(e) Histogram of preferred phases (red stars in d) for the entire population (n = 72). “

Methods:

- Both Spyking Circus and the KlustaKwik suite were used for spike-sorting: which data where used with which approach?

We have now updated the sentence under “data analysis”:

“Spike sorting was performed using Spyking Circus (Fig. 6) and Klustakwik (Figs. 7-8). “

- line after Equation 2: "fast" and "slow" are switched

Done!

Reviewer #2 (Remarks to the Author):

Radosevic et al attempt to tackle the interesting question of how a neuronal circuit is wired up, in either a sparse, convergent or dense convergent manner. The dual intracellular recordings and the combined multi-electrode array and intracellular recordings are technically demanding. The data analyses are extensive and equally challenging. While impressed with the data quality and analyses, the experimental design and interpretation of data are not very suitable for supporting the conclusions they have made.

Thank you for taking the time to review our work and provide helpful feedback.

The presumption of this type of connectivity analysis (Fig.1) is that when you are recording the postsynaptic neurons (Fig.5), they belong to the same functional group (module in this paper), and when you are recording from the presynaptic neuron (Fig.6-8), they are also from the same functional group, i.e. they form synapses on the neurons with the same physiological roles. Unfortunately, the spinal circuit is not organised in clear nuclei or layers per se, making selective recording from the same functional group of neurons using conventional dual intracellular recordings and multi-electrode array technically unachievable.

Yes the spinal cord is complex- Regarding your comment “...the spinal circuit is not organised in clear nuclei or layers per se...” do you have a reference addressing this? We would like to incorporate such a reference into the manuscript. The reason such a reference would be helpful, is that there is a general consensus within the spinal research field of modular organization in the spinal cord (not to mention the Rexed layers). See e.g. a recent review about the issue: “A recurrent theme, however, is that locomotor networks, whether they control swimming or

over-ground locomotion, are built around modules of rhythm- and pattern-generating networks that may be further functionally organized in to sub-modules. “ Kiehn Nat. Rev Neurosci 2016- And here from another influential report by the Arber lab: ”Our findings provide evidence for a discriminating anatomical basis of antagonistic circuits at the level of premotor interneurons, and point to synaptic input and developmental ontogeny as key factors in the establishment of circuits regulating motor behavioural dichotomy. “ Tripodi et al Nature 2011. Nevertheless, we do very much agree with you that the organisation of the spinal cord is by no means simple.

In theory, the more functionally diverse the recorded neurons are, the less likely a correlation will be found. To partially remedy this difficulty in meeting such presumptions in the chosen preparation, one could make many recordings and use post-experimental screening to identify possible neurons belonging to the same functional pool. This can be justified to some extent because rhythmic motor activity is mostly phasic. It is likely that neurons recorded with highly similar phase lags during scratching (i.e. recordings with high slow scale correlations) may belong to the same functional group. It also helps if the recordings are made from exactly the same location across preparations. This means the vast majority of existing recordings (Fig.5-8) need to be ditched for the analyses and a lot more recordings are going to be needed.

We agree that recording from the same location across preparations is helpful. We recorded from the same spinal segment across preparation of pairs within 300 um vicinity of each other, which was the closest our technique would allow. Our “post-experimental screening” was done by separation the slow and fast correlations. The slow correlation is a measure of the phase relationship, and the fast correlation is a measure of the shared synaptic input. We are glad that you see this strategy as a remedy. We find it difficult to imagine better methods.

Regarding “ditching” the data below $R=0.8$, we do not see the advantage of doing so. First, the points ($R < 0.8$) could still have contained partial common synaptic input. Since no one has done same or similar experiments previously, the issue is open. One could imagine a scenario where these neurons receive partial overlap in input, even though their phasic activity is not synchronized. If the cells receive few very strong and correlated input located at their somata, that would give high correlation on fast timescale, although there could be many weak

synaptic contacts located at distal dendrites with electrotonic filtering, thus inducing a weak or negative correlation on slow timescale.

Even if this scenario is unthinkable, these data could still work as a control. Their mean and variability give a sense of what can be expected by chance.

The anti-phase data ($R < -0.8$) also serves a purpose. The half center organization (fig. 1A) predicts that an inhibitory input should be expected to arrive in anti-phase with the antagonist module - if there is a dense divergent connectivity. This would result in values at -1, -1 in the plot. The fact that we do not see these negative values further substantiates the conclusion of sparse convergent architecture.

Take Fig.8 as just one example, the postsynaptic cell is the same neuron so there is no violation of the postsynaptic presumption. However, the random recording of many presynaptic neurons using the electrode array may sample neurons from many different presynaptic pools. This will inevitably lead to very low connectivity probability for inhibitory neurons.

Yes. We think we understand your point. This is confusing. Reviewer 1 also brought up a similar point. The point is that in Fig. 8 we do not know if the established connections are from the common drive network (yellow in Fig. 1) or recurrent connectivity within the receiver network (blue in fig. 1). However, the risk of recording from the source network by accident is very small in the dense divergent scenario (Fig. 1b). We have now clarified this in the text in the introduction:

“First, we utilize dual intracellular recordings to assess the strength of synaptic correlations, in particular for pairs belonging to the same module. The modular–association is based on two issues: 1) motor neuron pairs in close vicinity and with same slow phase 2) interneurons also in close vicinity that have same phase are assumed to belong to same module and receive common drive. This assumption is based on the consensus view that the common–drive network is small compared with the receiver network (hence the term ‘common’) and therefore the risk of randomly recording from one of the source–network neurons is equally small. Next, we use multi–electrode arrays to measure population activity to determine the pairwise spike–spike correlation as an alternative indicator for shared synaptic input, under same

assumptions.“

Regarding the issue of connectivity: It is true that the sampling from different pools will inevitably lead to low connectivity- Nevertheless, it is difficult to imagine how to do it otherwise. It is difficult to group the cells among which we would like to measure the connectivity. By which criteria should we select our group of neuron?

If we define a group of neurons by selecting the ones receiving synapse from neurons with same function - then it is circular argument: The relevant neurons are the ones that has a connection, and therefore the connectivity would be 100% within this group per definition. In this context it is also interesting, that there are 3 inhibitory units out of phase, and 1 inhibitory in phase with the motor cycle (fig. 8b) - which is a sign that the phase itself is not a reliable indicator of connectivity. So how should one group the neurons? To avoid the above caveat the connectivity probability should be calculated indiscriminately between pairs, at least as a first approach.

Why should the correlation values for slow and fast scale signals be locked in a linear manner (Fig.5a, 7c)? Such relation may be exponential, for example. Therefore it will be interesting to analyse the points where correlation of slow scale signal is >0.8 and carry out some data transformation to re-examine the relation, rather than using a unity line which covers from -1 to 1.

The most parsimonious relationship to assume when nothing else is known is a linear relationship. Yes, it could also be an exponential- keeping in mind that it should also go negative for the anti-phase pairs, in which case it would be two truncated exponential with opposite signs. We are unsure what you mean by data transformation and reexamination- However, we have performed an exponential fit (see below) to the pairwise recordings (all, red curve) and to only the $R > 0.8$ (blue curve, below). There is a small upward curvature due to the 3 out of 17 points above the confidence interval. The root-mean-square error is larger for the blue line (RMSE=0.090) than it is for the red curve (RMSE=0.076) We are unsure what to conclude from this fit. Regardless, the 3 points are above the confidence limit, they still represent a rather weak correlation ($R^2 = 10\%$) especially compare with the same cell recordings. We have added a sentence in the result, reporting these numbers:

“When choosing slow correlation > 0.8 there was $n=3/17$ points (17.6%) above confidence limit.”

Fig.1 State: grey symbols are local premotor neurons and white ones are motoneurons (MN)

We have now modified the caption sentence to the following:
“...to flexor- and extensor-related neurons in the spinal cord (blue shaded region with local premotor neurons) including reciprocal inhibition (IN).“

Last paragraph on p2, citation of Fig.1b, Fig.1c is incorrect.

We are unsure about this point. When carefully reading page 2, we think it is the following sentence. We do not think there is a mistake in the text. Nevertheless, we have added a referral “(yellow)” to the particular population in in the context to increase clarity:

“A given number of arriving synaptic connections to a group of neurons can either be provided by small population neurons (yellow) with many axon collaterals (Fig. 1b), i.e., a dense/divergent connectivity, or a large population (yellow) with few axon collaterals, i.e., a sparse/convergent connectivity (Fig. 1c). “

Fig.2, n=20 cells but the histogram suggests more cells.

Yes it is a bit confusing. The n=20 refers to the source population, whereas the correlation matrix in Fig 2C (colored square) refers to a subset of the receiver population (30 in this case). The histogram on the left, is the distribution of pairwise correlation, which since it is the combinatorial measure, is a much larger number. We have added the word “source cells” in the caption to help clarity:
“(n = 20 source cells) “

Fig.3d, label IN, MN instead of neuron3, neuron 4.

Yes. Because of the close proximity (<300 um) of the neurons we do not know with absolute certainty which of the recorded neuron belong to the identified MN and IN. But we know that the pair is a IN-MN pair. We suggest leaving the Neuron 1 and neuron 2 because of this.

Fig.4, explain “shuffled data correlation”.

We have now added the following sentences in the method section:

“A shuffled correlation was used for comparison, in which any causal correlation was eliminated by randomly shifting one trace while leaving the other trace intact. In this way, we could establish the correlation expected purely by chance. The shuffled recording was constructed by shifting the trace in time, in a region that had similar statistics of synaptic intensity. The shift was chosen randomly from trial to trial between 500 ms to 2 sec. ”

Reviewer #3 (Remarks to the Author):

This interesting paper considers the broad question of correlations in spinal motor neurons and interneurons during rhythmic activity and its implications for network connectivity and mechanisms. The preparation used is that of the turtle spinal cord, during hindlimb scratching, which offers a number of unique advantages, as a result of which the results reported here are the first of their kind in many respects. The work is rigorous and well-presented.

Thank you for taking the time to read our manuscript and provide constructive comments.

Before getting to detailed comments/questions, I want to mention that as someone not active in the subfield of motor circuits, some of my comments are possibly addressed by short remarks and appropriate citations, that would also serve the non-aficionado reader.

Comments/questions:

1. The authors state that their results are consistent either with the possibility of sparse convergent connectivity and/or a mechanism that performs active decorrelation. The title however only reflects the former possibility. The authors should consider either hedging the title a bit to accommodate the latter possibility or perhaps have a more general title that does not mention either. In short, my concern is that if future work in fact shows evidence for the active decorrelation, the title would not age well.

Yes. We chose this title to reflect the network property which we consider most likely. Further it is also compatible with both the scenario of active decorrelation as well as the conventional wisdom of spinal

motor networks. The conventional wisdom regarding spinal circuits does not include recurrent feedback inhibition, and therefore this property would represent a substantial revision of the current models. We do not think that the data presented in this study really supports such a radical revision of the established models, we also do not wish to explicitly make this the scope of our paper. Active decorrelation, should be presented as possible explanation for inspiration, but we suggest not including it in the title at this point. Also, the new addition to the issue (figure 9 and 10 in the revision) indicates that the active decorrelation is spike rate dependent, and therefore the correlation should wax and wane though the motor cycles. We do not see this property clearly in the data (although it could be in a high firing regime), so we are now less certain about active decorrelation.

2. The experimental preparation requires decapitation and removal of muscles that limit proprioceptive feedback. It would be useful to see a short discussion of how experimental results so obtained might differ from that of the same circuits in a fully intact animal.

Yes. It is believed that the brain acts in part to inhibit lower level motor programs and reflexes. Hence, the removal of the head strengthens the motor response to sensory input - like somatic touch in our case. Further action of supraspinal centers is primarily midbrain, reticular formation, motor cortex, basal ganglia and cerebellum, which have to do with planning, timing of execution and correction in coordination of movement. All these are not so relevant in the simple elements of the rhythmic motor behaviors. The proprioceptive feedback usually increase the motor response, and therefore the removal reduce the network activity. The point of removing this feedback is to tease out the centrally generated motor behavior and the circuitry without peripheral interference. We have modified the text in the results section in the manuscript, and stress that supraspinal input often prevents the spinal reflex:

“To avoid the confounding factor of supraspinal input, the interference with and prevention of the scratching response to sensory stimulation, the turtles were spinalized. The muscles were removed to limit proprioceptive feedback and increase mechanical stability, while leaving the cutaneous sensation intact. “

3. The model, as I understand it, considers the case where the source

neurons are themselves uncorrelated. It would be useful to see analysis (described possibly in Supplementary) of the case where the source neurons are also correlated, and how that affects the plots. of Fig 2.

Yes. If all or a portion of the source neurons were correlated, it would be identical (for the receiver cells) to the situation of action potentials originating from the same neurons, with more synaptic connections, i.e. a denser network. This would be relevant if our data showed very high correlation. We would face the dilemma of not being able to distinguish these two situations, i.e. large correlated population with sparse connectivity versus a small uncorrelated population with massive divergence. Fortunately, our data show the opposite, i.e. very low, or no correlation.

4. Apropos Fig 3(e), are all neuron pairs oscillating in the same frequency? Were pairs of non-commensurate frequency found? I don't recall seeing this being addressed.

Yes. The activity of all neurons have basically two parts: a rhythmic activity all with same frequency but different phase lags, and a slow increase and then a decrease following the bout. We have not seen bursting frequencies that is different than the motor rhythm, neither in our data nor in the literature. We have added a note in a sentence about the commensurate frequencies in:

“In fact, all of the neuronal pairs had commensurate oscillations with various phase lags (Fig. 3e) regardless of their close physical location...”

5. To what extent is the lack of fast-timescale correlations due to physiological variability between cells? As I understand it, the present model has cells with identical parameters. Some modeling to address this would be insightful. For example, from the data, one question could be: Is the ratio of spike counts across peaks of the slow timescale between neuron pairs somewhat preserved?

Yes. It is a difficult issue to properly address. The only relevant physiological variability (as we see it) that could be important for the difference in timing of synaptic input, i.e. could “decorrelate” correlated input somehow, is electrotonic filtering. We cannot really think of other relevant physiological parameters. We intended to address this issue in our experiments by performing pairwise recordings from the same

neuron. Hence we were able to see how much variability there is between recordings at different locations of neurons. Although, we only had only 5 such pairwise recordings, they all had strikingly close synaptic activities. This is the reason we do not attribute electrotonic filtering (and variability thereof) much power as explanation for the lack of correlations.

6. Is there a reason the distribution of slow time scale correlations is markedly different between Fig 7d and Fig 5b?

Yes - it's puzzling. We think there are three reasons for the disparity in distributions: 1) The intracellular experiments are more direct in identifying relevant rhythmic neurons (Fig. 5b), and perhaps do not represent the arrhythmic neurons so well. 2) The multi-electrode arrays record from all neurons indiscriminately of their relationship with motor, and hence the pairwise correlation cluster around zero. For the same reason we performed the Rayleigh test for rhythmicity to exclude some of the units that were arrhythmic. Nevertheless, there are still neurons that have very weak rhythmicity in the population, and these give a peak at zero. 3) In general, the population firing rate distribution is lognormal (see Petersen and Berg 2016) - Many neurons, even the ones directly related to the rhythm, spike at such low rate, that the firing rate estimates are poorly modulated by the rhythm. If a pair of neurons - even if their are directly related to the behavior - only fire a couple of time during a motor bout, it is difficult to establish their rhythm and phase. All these pairs will tend to cluster around zero.

7. The video abstract is nicely done and provides a good, quick, first overview of the work. I would suggest retaining it.

Thank you- The journal is welcome to use it as supplementary information along with the paper. We can polish it if requested.

8. I appreciate the authors starting with the model that forms the backdrop for the rest of the paper.

Yes. We will keep the model in the start of the paper. Reviewer 1 has asked for an elaborate that includes recurrent connectivity in the receiver population. That has been included in the end of the paper, i.e. figures 9-10 and the last section in the results.

Minor comments:

1. Page 5, last paragraph: “separation” is misspelled as “seperation”.

Done!

2. Fig 6 legend: Panel (e) hasn't been explicitly mentioned.

Yes. Reviewer 1 had some observation. We have now added the following in the caption:

“(e) Histogram of preferred phases (red stars in d) for the entire population ($n = 72$). “

In closing, this is a strong paper with comprehensive data and interesting results that will spur further theory and experiment. I would recommend that the paper be accepted with a minor revision.

Thank you for the constructive feedback-

Reviewers' Comments:

Reviewer #1:

Remarks to the Author:

The authors have done an excellent job with the revised manuscript. It is much clearer throughout; in particular, it is clear how the experiments are guided by, and informing the conclusions from, the initial sender/receiver models. The authors have done a particularly excellent job of following up my suggestion of looking at active decorrelation in their models and data - there are now valuable additional results here on the complex, non-monotonic impact of feedforward recurrent inhibition on correlations between neurons in a target population (Fig 9, Fig 10). I thought the analysis of rate-dependent correlations in Fig 10 was very creative, by comparing the waxing and waning of the pairwise correlations to the Vm as a proxy for firing rate. By ending on the final suggestion that perhaps the spinal network does not contain active decorrelation, the data here should spur further theoretical work on what kind of network would support the decoupling of rate and correlation in Fig 10d.

Mark Humphries

Reviewer #2:

Remarks to the Author:

This reviewer is fully aware that more and more evidence suggests that the spinal circuit is organized in modules. However, this does not change the fact that your recording methods are sampling heterogenous groups of neurons, partially violating the assumptions for sparsity estimation. The task for the research question could be a lot simpler, to name an example, by investigating the connections between the inferior olive neurons and Purkinje cells. The difficulty of using an adult spinal preparation, which has been recognised by the workers in their response, is inherently immense. In this sense, the efforts and data the workers have attempted do deserve appreciation perhaps without overly harsh treatment.

While this reviewer does not see any available recording methods that can overcome this inherent difficulty, I do think some separate evaluation of data points divided by $R=0.8$ is one approach not to make the said difficulty go away, but to recognise it and implement more caution in data analyses and drawing conclusions. As the workers suggested, they could separately analyse the data and use data with $R<0.8$ as controls. At the same time, clear discussions need to be made to relate your conclusions to the heterogenous nature of neuronal grouping in your samples and your assumptions for sparsity estimation.

All other points have been satisfactorily addressed.

Reviewer #3:

Remarks to the Author:

The authors have addressed all of the comments from my initial review.

Apropos response to my Comment #6 ("6. Is there a reason the distribution of slow time scale correlations is markedly different between Fig 7d and Fig 5b?"), it would be good to include a short paragraph addressing this issue perhaps in the text corresponding to Fig 7 or in a sub-section in Supplementary. This is an issue that might occur to other readers or researchers building on top of

this work and I always find that it serves Science to include observations that are puzzling (in the authors' words).

Barring this one point, which is easily addressed, I am satisfied with the author response.

In closing, I want to re-iterate that this is a strong paper with comprehensive experiments and several results that are the first of their kind. I would therefore recommend it for acceptance, without further delay.

Reviewers' comments:

Reviewer #1 (Remarks to the Author):

The authors have done an excellent job with the revised manuscript. It is much clearer throughout; in particular, it is clear how the experiments are guided by, and informing the conclusions from, the initial sender/receiver models. The authors have done a particularly excellent job of following up my suggestion of looking at active decorrelation in their models and data - there are now valuable additional results here on the complex, non-monotonic impact of feedforward recurrent inhibition on correlations between neurons in a target population (Fig 9, Fig 10). I thought the analysis of rate-dependent correlations in Fig 10 was very creative, by comparing the waxing and waning of the pairwise correlations to the Vm as a proxy for firing rate. By ending on the final suggestion that perhaps the spinal network does not contain active decorrelation, the data here should spur further theoretical work on what kind of network would support the decoupling of rate and correlation in Fig 10d.

Mark Humphries

Again, thank you for taking the time to review our manuscript and providing valuable input.

Reviewer #2 (Remarks to the Author):

This reviewer is fully aware that more and more evidence suggests that the spinal circuit is organized in modules. However, this does not change the fact that your recording methods are sampling heterogeneous groups of neurons, partially violating the assumptions for sparsity estimation. The task for the research question could be a lot simpler, to name an example, by investigating the connections between the inferior olive neurons and Purkinje cells. The difficulty of using an adult spinal preparation, which has been recognised by the workers in their response, is inherently immense. In this sense, the efforts and data the workers have attempted do deserve appreciation perhaps without overly harsh treatment.

While this reviewer does not see any available recording methods that can overcome this inherent difficulty, I do think some separate evaluation of data points divided by $R=0.8$ is one approach not to make the said difficulty go away, but to recognise it and implement more caution in data analyses and drawing conclusions. As the workers suggested, they could separately analyse the data and use data with $R<0.8$ as controls.

We have now performed and provided a statistical analysis of the pairwise correlations. The data was binned for the slow correlation with same width (0.2) from -1 to +1, and performed a pairwise comparison of neighboring distributions within these bins. The analysis is provided in a supplementary figure 1. The particular bin of interest, the $R>0.8$ as you pointed out, is compared with the bin of 0.6-0.8. A two sample t-test was unable to reject the null hypothesis of equal mean of the fast correlation, although the statistical power of the comparison was not too great (0.21). This should help the reader judge the data sample and verify the conclusions of the paper. Please find supplementary figures and text associated with this analysis. We have added the following sentence in the results:

“When choosing slow correlation > 0.8 there was $n=3/17$ points (17.6%) above confidence limit. Nevertheless, the fast correlation in this group was not statistically different from the group with slow correlation below 0.8 (Supplementary figure 1). “

At the same time, clear discussions need to be made to relate your conclusions to the heterogenous nature of neuronal grouping in your samples and your assumptions for sparsity estimation.

We have written the following section about heterogeneity and added it in the discussion:

“Heterogeneity of the neuronal population

The population of neurons investigated in the current study was classified according to their electrical activity, which is agnostic towards their genetic identity. Although, spinal interneurons are often categorized according to their cellular lineage, it is unclear if such categorization would have a simple relationship with the electrical activity. There is no indication of exclusive connectivity between interneuron subtypes and

the various motor pools. Multiple clades of interneurons can innervate a single motor pool, and the same clade can innervate multiple motor pools.⁵⁴ Further, it is uncertain if cells of same subtype should have a preference of being interconnected, rather than being connected to cells of other origin. A recent investigation using intracellular recordings from pairs of interneurons with the *shox2* transcription factor demonstrated the internal connectivity among those interneurons to be sparse.⁵⁵ These interneurons together with those expressing the Hb9 transcription factors have been suggested to be responsible for rhythm generation^{56,57,58} and could represent the source neurons in the feedforward network presented here. Nevertheless, the motor circuitry is likely to be made up of a heterogeneous population with various electrical activities interconnected in a complex manner, which remains to be unraveled.⁴⁰ “

All other points have been satisfactorily addressed.

Thank you for taking the time to review our work and providing important comments.

Reviewer #3 (Remarks to the Author):

The authors have addressed all of the comments from my initial review.

Apropos response to my Comment #6 ("6. Is there a reason the distribution of slow time scale correlations is markedly different between Fig 7d and Fig 5b?"), it would be good to include a short paragraph addressing this issue perhaps in the text corresponding to Fig 7 or in a sub-section in Supplementary. This is an issue that might occur to other readers or researchers building on top of this work and I always find that it serves Science to include observations that are puzzling (in the authors' words).

Yes this is important to have in the manuscript. We have now added a section in the supplementary with a similar explanation:

“Distribution of slow correlation for intracellular versus extracellular data
The distribution of slow timescale correlations is qualitatively different for

the pairwise intracellular recordings compared with the spike rate correlations of the extracellularly acquired pairs (cf. Fig 5c and 7d). There are likely three reasons for this disparity in the experimental data: 1) The intracellular recordings were directly aimed at identifying relevant rhythmic neurons (Fig. 5b), while not necessarily representing the arrhythmic neurons. 2) The multi-electrode arrays recorded neurons indiscriminately of their relationship with motor activity, and hence the pairwise correlation representing the weakly rhythmic activity cluster around zero. For the same reason we performed the Rayleigh test for rhythmicity to exclude some of the units that were arrhythmic. Nevertheless, many units with weak rhythmicity remained in the population, and these give a peak at zero. 3) The population firing rate distribution is lognormal (Petersen and Berg 2016). Hence, many neurons, even the ones directly related to the rhythm, spike at such a low rate, that the firing rate estimates are poorly modulated by the rhythm. If a pair of neurons - even if their are directly related to the behavior - only fire a couple of time during a motor bout, it is difficult to establish their rhythm and phase. All these pairs make out most of the mass in the distribution around zero in the extracellular data.”

Barring this one point, which is easily addressed, I am satisfied with the author response.

In closing, I want to re-iterate that this is a strong paper with comprehensive experiments and several results that are the first of their kind. I would therefore recommend it for acceptance, without further delay.

Thank you for important comments and taking the time to review our manuscript.